# DAPD: Dependency-Aware Parallel Decoding via Attention for Diffusion LLMs

**Bumjun Kim** [* 1]  **Dongjae Jeon** [* 2]  **Moongyu Jeon** [* 1]  **Albert No** [1]

## Abstract

Parallel decoding for Diffusion LLMs (dLLMs) is difficult because each denoising step provides only token-wise marginal distributions, while unmasking multiple tokens simultaneously requires accounting for inter-token dependencies. We propose **D**ependency-**A**ware **P**arallel **D**ecoding (**DAPD**), a simple, training-free decoding method that uses self-attention to induce a conditional dependency graph over masked tokens. At each iteration, edges in this graph capture strong token interactions, while non-edges indicate weak dependence. Parallel decoding is then reduced to selecting an independent set on the graph and unmasking the selected tokens in parallel. This avoids co-updating strongly coupled tokens without auxiliary models or retraining. Experiments on LLaDA and Dream show that DAPD improves the accuracy–steps trade-off over existing methods and enables more globally distributed parallel updates that better exploit the any-order generation capability of dLLMs. The project is available at https://ai-isl.github.io/dapd

## 1. Introduction

Diffusion models have achieved remarkable success in continuous domains such as image and audio generation (Sohl-Dickstein et al., 2015; Song & Ermon, 2019; Ho et al., 2020; Song et al., 2021), and have recently been extended to discrete data (Hoogeboom et al., 2021; Austin et al., 2021a; Campbell et al., 2022; Lou et al., 2024). In particular, Masked Diffusion Models (MDMs) (Sahoo et al., 2024; Shi et al., 2024; Ou et al., 2025) have emerged as a promising alternative to autoregressive (AR) models (Radford et al., 2018; 2019). With large-scale models such as LLaDA (Nie et al., 2025) and Dream (Ye et al., 2025), masked diffusion-based LLMs (dLLMs) demonstrate strong scalability and competitive language modeling performance.

Parallel decoding, the ability to generate multiple tokens simultaneously, is often highlighted as a key advantage of dLLMs (Nie et al., 2025; Ye et al., 2025), offering the potential to significantly reduce inference latency compared to the serial nature of AR generation. However, realizing this potential is challenging: dLLMs are typically trained to model only token-wise conditional marginals at masked positions, and marginals alone do not determine the true joint distribution. Consequently, independently sampling multiple tokens from these marginals ignores their inter-dependencies, leading to *joint–marginal mismatch* and often producing outputs that are locally plausible yet globally inconsistent (Ben-Hamu et al., 2025; Wu et al., 2026).

Existing training-free parallel decoding strategies (Wu et al., 2026; Ben-Hamu et al., 2025; Kim et al., 2025b) often sidestep joint dependencies by relying on marginal-based signals such as confidence scores (Chang et al., 2022; Kim et al., 2025a). While such methods filter out uncertain tokens, they do not account for interactions between masked positions, leaving the joint–marginal mismatch unresolved when multiple confident yet dependent tokens are decoded in parallel. Parallel decoding methods with additional training or auxiliary planners have also been explored (Israel et al., 2025; Bao et al., 2026; Chen et al., 2026), but they often incur extra complexity or rely on surrogate objectives.

In this work, we aim to capture joint dependencies to address the joint–marginal mismatch, utilizing the model's self-attention as a proxy for conditional independence. Building on this insight, we model the interactions among masked tokens as a *Markov Random Field (MRF)*. We use attention to estimate token dependencies and represent them as a graph, where edges denote significant interactions. We empirically validate this representation through controlled experiments on synthetic data, confirming that attention signals reliably capture the ground-truth interaction structure. This graph serves as a structural guide for identifying which tokens can be safely decoded in parallel.

We then cast parallel decoding through the lens of graph coloring, a classical abstraction for scheduling mutually conflicting items: given a graph where edges connect pairs that should not be processed together, each color class corre-

---

[*]Equal contribution  [1]Department of Artificial Intelligence, Yonsei University, Seoul, Korea [2]Department of Computer Science, Yonsei University, Seoul, Korea. Correspondence to: Albert No <albertno@yonsei.ac.kr>.

*Proceedings of the 43rd International Conference on Machine Learning*, Seoul, South Korea. PMLR 306, 2026. Copyright 2026 by the author(s).

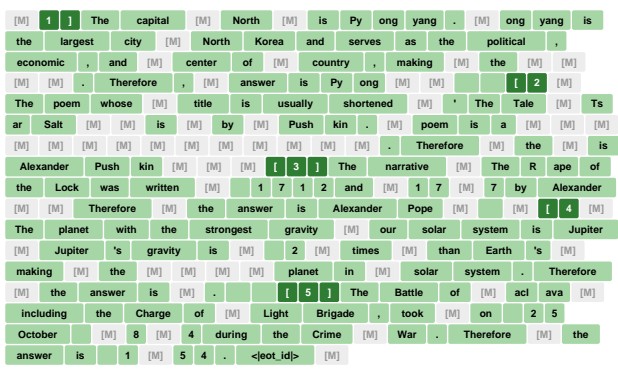

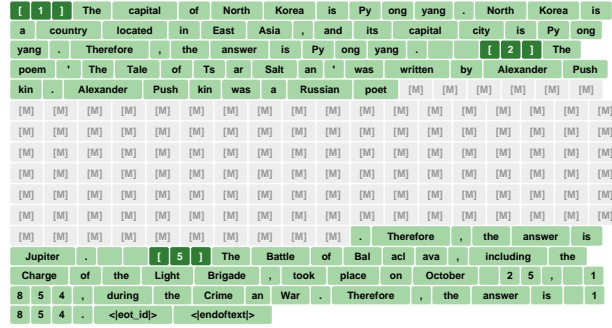

| (a) DAPD (Ours) | (b) Fast-dLLM |

*Figure 1.* Visualization of unmasking trajectories for DAPD (ours) and Fast-dLLM during the initial 50% of decoding steps. While DAPD processes independent questions simultaneously to maximize global parallelism, Fast-dLLM exhibits a highly sequential unmasking pattern. Other baselines demonstrate similar sequential behaviors (see Figs. 9 to 12). Detailed discussion is provided in Sec. 6.

sponds to an independent set that can be handled in parallel. Under this view, parallel decoding in dLLMs becomes a dynamic coloring problem, where masked tokens are nodes and attention-induced couplings define edges. Selecting a set of tokens to unmask in parallel thus reduces to choosing an (approximately) independent set on the current graph, which serves as a linguistically motivated relaxation of joint independence and helps mitigate joint–marginal mismatch. We implement this idea using a Welsh–Powell-inspired heuristic (Welsh & Powell, 1967) that prioritizes high-degree "hub" tokens, simplifying the residual graph across decoding steps. Across benchmarks on LLaDA (Nie et al., 2025) and Dream (Ye et al., 2025), **DAPD** consistently achieves a better accuracy–steps trade-off than state-of-the-art baselines.

DAPD is training-free, relying only on model-internal signals available at inference time to guide parallel unmasking, without any additional training cost or auxiliary models. Beyond this simplicity, we observe a qualitative difference in the resulting decoding dynamics. Baseline strategies driven primarily by marginal confidence tend to unmask tokens in contiguous clusters, effectively behaving like two-sided autoregressive decoding that expands inward from the sequence boundaries and underutilizes global context. In contrast, DAPD generates tokens in a spatially dispersed manner, prioritizing independent regions across the entire sequence before filling in the gaps. This highlights a key advantage of discrete diffusion: leveraging bidirectional context for non-sequential, global generation, and suggests global parallelism as a natural decoding paradigm for dLLMs.

## 2. Preliminaries

### 2.1. Masked Diffusion Models

In Masked Diffusion Models (MDMs), the diffusion process is implemented through progressive token masking. Specifically, a subset of tokens in a sequence $\mathbf{x} = x^1 \ldots x^L \in \mathcal{V}^L$

is replaced with a special mask token $[\mathbf{M}]$, and the model learns to recover the original tokens (Austin et al., 2021a; Lou et al., 2024). For a uniformly selected time $t \in [0, 1]$, each token in the original sequence $\mathbf{x}_0 = x_0^1 \ldots x_0^L$ is independently masked with probability $t$ to produce a corrupted sequence $\mathbf{x}_t$. The model is trained to predict the original token at each masked position by learning the conditional probability $p_\theta(x_0^i | \mathbf{x}_t)$, using the following weighted cross-entropy loss. This loss serves as a variational upper bound on the negative log-likelihood $-\log p_\theta(\mathbf{x}_0)$ (Sahoo et al., 2024; Shi et al., 2024; Ou et al., 2025):

$$\mathcal{L}_{\text{MDM}}(\mathbf{x}_0; \theta) = -\mathbb{E}_{t, \mathbf{x}_t} \left[ \frac{1}{t} \sum_{i:x_t^i=[\mathbf{M}]} \log p_\theta(x_0^i | \mathbf{x}_t) \right].$$

After training, the model provides token-wise conditional distributions $p_\theta(x^i | \mathbf{x})$ for all positions $i \in \{1, \ldots, L\}$ and masked sequences $\mathbf{x} \in \overline{\mathcal{V}}^L$, where $\overline{\mathcal{V}} = \mathcal{V} \cup \{[\mathbf{M}]\}$. These conditionals generate a full sequence $\mathbf{x}_0$ by iteratively unmasking tokens from the fully masked input $\mathbf{x}_1 = [\mathbf{M}] \cdots [\mathbf{M}]$. According to the factorization

$$p(\mathbf{x}_0) = \prod_{i=1}^{L} p(x^{\pi(i)} | \mathbf{x}^{\pi(<i)}),$$

unmasking the sequence according to any permutation $\pi$ on $\{1, \ldots, L\}$ theoretically yields samples from the same distribution. In practice, however, the choice of unmasking order is known to have a significant impact on generation quality (Zheng et al., 2024; Kim et al., 2025a). A common and effective strategy is confidence-based sampling (Chang et al., 2022), which prioritizes the masked position whose predictive distribution has the largest maximum probability. Intuitively, by unmasking tokens with the lowest uncertainty first, this approach prevents error propagation and ensures that subsequent predictions rely on a trustworthy context.

More recently, many MDM variants have adopted a semi-autoregressive block generation scheme (Arriola et al., 2025; Wu et al., 2026), in which a sequence is partitioned into fixed-length blocks. Within each block, tokens are generated using diffusion, while blocks are processed sequentially in a left-to-right order. This formulation leverages the inherent left-to-right dependency of natural language across blocks and enables block-level KV caching during inference.

Recent work has shown that MDMs can be successfully scaled to large language models, giving rise to Diffusion Large Language Models (dLLMs) (Khanna et al., 2025). These include models trained either from scratch or via autoregressive initialization, such as LLaDA (Nie et al., 2025) and Dream (Ye et al., 2025).

### 2.2. Parallel Decoding of dLLMs

**Challenge of Parallel Decoding.** Parallel decoding is often highlighted as a key advantage of dLLMs, allowing multiple tokens to be generated simultaneously at each denoising step. The computational cost of dLLMs is dominated by the number of model forward passes, often referred to as function evaluations (NFE). Since each decoding step requires exactly one forward pass to update the sequence, the total NFE is equivalent to the number of decoding steps. Thus, minimizing the step count is crucial for efficiency. By unmasking multiple tokens within a single step, parallel decoding can significantly reduce this cost.

However, dLLMs are trained to model conditional marginal distributions at each masked position. As a result, unmasking multiple tokens simultaneously can yield errors due to mismatch between the true joint distribution and the product of marginals (Ben-Hamu et al., 2025; Wu et al., 2026):

$$p(x^{i_1}, \ldots, x^{i_k}|\mathbf{x}) \neq \prod_{j=1}^{k} p(x^{i_j}|\mathbf{x}). \tag{1}$$

When multiple tokens are decoded in parallel, this discrepancy can accumulate, as dependencies among the tokens are not fully accounted for.

As an illustrative example, consider the prompt "The capital of [**M**] is [**M**]" (Song & Zhou, 2025). The model may assign a high marginal probability to "France" at the first position and to "London" at the second. Although each prediction is plausible in isolation, sampling both simultaneously can yield an inconsistent pair, showing that ignoring dependencies between masked positions can lead to invalid outputs.

**Training-free parallel decoding for dLLMs.** As discussed above, a central difficulty of parallel unmasking in dLLMs is the *joint–marginal mismatch* (Eq. (1)), which arises when masked positions are interdependent. Most training-free samplers therefore focus on designing crite-ria to select positions that can be safely decoded together, typically using marginal signals available at inference time.

We consider several representative training-free strategies as baselines. The most direct approach ranks masked positions by marginal confidence and decodes the top-$k$ ranked positions in parallel (Nie et al., 2025). **EB-Sampler** (Ben-Hamu et al., 2025) instead selects the subset of positions that satisfies an entropy-based bound, aiming to control the risk of decoding dependent tokens together. **Fast-dLLM** (Wu et al., 2026) applies selective parallelism by unmasking only positions whose confidence exceeds a fixed threshold. **KLASS** (Kim et al., 2025b) further augments confidence with a stability signal defined by the token-wise KL divergence between consecutive denoising steps, and parallel-unmasks tokens that are both confident and stable.

Beyond the above training-free methods, there exist parallel decoding approaches that rely on additional training, auxiliary models, or modified strategies (Israel et al., 2025; Bao et al., 2026; Chen et al., 2026), which we do not pursue here. These methods handle inter-token coupling more explicitly, but incur extra cost and may depart from the ELBO-based formulation via surrogate objectives, auxiliary targets, or training-inference mismatches.

## 3. Modeling Dependencies via Graphs

Departing from existing marginal-based strategies that neglect cross-position interactions (Sec. 2.2), we introduce a method that leverages the model's internal self-attention mechanism to capture the dependency structure of the masked tokens. We formulate this structure as a *Markov Random Field (MRF)*, where the attention mechanism defines the conditional independence properties between variables. Within this probabilistic framework, we reduce parallel decoding to selecting an independent set of the induced MRF.

### 3.1. Attention-Induced Markov Random Fields

We model the interactions among masked positions at each step $t$ as an MRF, which explicitly characterizes the conditional independence structure of the masked tokens.

**Definition 3.1** (Markov Random Field). A collection of random variables $X = \{X_v\}_{v \in V}$ constitutes a *Markov Random Field (MRF)* with respect to an undirected graph $G = (V, E)$ if it satisfies the *pairwise Markov property*: for any two non-adjacent nodes $u, v \in V$ (i.e., $(u, v) \notin E$), the variables $X_u$ and $X_v$ are conditionally independent given all other variables $X_{V \setminus \{u,v\}}$.

*Remark* 3.2. Another common way to define an MRF is via a clique-factorized form. Under positivity (i.e., $p(x) > 0$ for all $x$), this definition is equivalent to the pairwise Markov property with respect to $G$ (Koller & Friedman, 2009).

Intuitively, the absence of an edge means that, once the global context is given, the two variables no longer provide information about each other.

**Graph Construction in dLLM Decoding.** We construct an MRF induced by a pretrained dLLM. At each decoding step, masked positions can be viewed as random variables, and our goal is to characterize their dependencies by constructing the corresponding MRF graph. We use attention to define the edge structure, treating stronger attention as indicating a stronger dependency.

Let $V_t = \{i \in \{1, \ldots, L\} : x_t^i = [\mathbf{M}]\}$ denote the set of masked indices at step $t$. We construct a graph $G_t = (V_t, E_t)$, where the edge set $E_t$ is determined using the model's self-attention maps. More precisely, we define a symmetric *edge score* $s_{ij}$ to quantify the interaction strength between positions $i$ and $j$:

$$s_{ij} := \frac{1}{2}(a_{ij} + a_{ji}),$$

where $a_{ij}$ represents the attention weight from $i$ to $j$ (the choice of layer and head is specified in Secs. 3.2 and 4.3). We interpret the magnitude of this score as a proxy for the strength of conditional dependence, thereby aligning this structure with the edge set $E_t$.

Using a threshold $\tau_t \geq 0$, we establish the edges as follows:

$$(i, j) \in E_t \quad \Longleftrightarrow \quad s_{ij} > \tau_t.$$

Under this construction, an edge ($s_{ij} > \tau_t$) represents a considerable dependency, while a non-edge ($s_{ij} \leq \tau_t$) indicates that the interaction is negligible.

**Justification: Attention as Conditional Independence.** We justify our attention-based edge construction by relating low attention to conditional independence. In a Transformer, negligible attention between positions $i$ and $j$ indicates that the model places little weight on the representation at $j$ when forming the contextualized representation used to predict the token at $i$. As a result, once we condition on the remaining context $X_{V_t \setminus \{i,j\}}$, the model's predictive distribution for $X_i$ becomes effectively invariant to whether $X_j$ is revealed. This intuition is captured by the approximation

$$p_\theta\big(X_i \mid X_{V_t \setminus \{i\}}\big) \approx p_\theta\big(X_i \mid X_{V_t \setminus \{i,j\}}\big),$$

Namely, $X_i$ and $X_j$ are approximately conditionally independent given the remaining variables $X_{V_t \setminus \{i,j\}}$. Therefore, defining non-edges via low attention is consistent with the local Markov property underlying MRFs.

### 3.2. Validation of Graph Construction

We empirically validate our MRF construction in a controlled toy setting to verify whether attention signals can

effectively capture the underlying conditional independence structure and distinguish between graph edges. Experimental details are provided in App. B.



*Figure 2.* Illustration of synthetic MRF dataset.

**Setup.** We train simple masked diffusion models (MDMs; RADD; Ou et al. (2025)) on a synthetic dataset composed of length-9 sequences $(X_1, \ldots, X_5, Y_1, \ldots, Y_4)$. The variables $(X_1, \ldots, X_5)$ are sampled independently from the discrete uniform distribution on $\{0, 1, 2\}$, and the remaining tokens are deterministically defined by

$$Y_i := (X_i + X_{i+1}) \bmod 3, \qquad i = 1, 2, 3, 4.$$

This construction induces four local constraints, each involving the triple $(X_i, X_{i+1}, Y_i)$. Accordingly, we represent the ground-truth MRF structure as the union of four triangles:

$$\{X_i, X_{i+1}, Y_i\}, \qquad i = 1, 2, 3, 4.$$

Although $X_i$ and $X_{i+1}$ are marginally independent, the constraint $Y_i \equiv X_i + X_{i+1}$ induces conditional dependence given $Y_i$, necessitating a fully connected triangle for each triple. Thus, as shown in Fig. 2, the ground-truth MRF consists exclusively of four triangles, with no connections between variables not involved in a common constraint.

*Table 1.* Metrics for edge detection and degree estimation. AUC and edge/non-edge attention ratio measure edge detection, and OVR measures degree estimation. Results are averaged over 100 random sampling paths and all decoding steps. Further step-wise and layer-choice results are in Tabs. 9 and 10.

| Edge Detection | | Degree Estimation |
|---|---|---|
| AUC ↑ | Ratio (Edge/Non-edge) ↑ | OVR ↓ |
| 0.928 | 2.204 | 0.04 |

**Edge Score.** For each pair of positions $(i, j)$, we compute the edge score $s_{ij}$ by averaging the attention weights over all heads and the top two layers of the eight-layer Transformer. We focus on upper layers because they tend to integrate information propagated across the network, capturing more global dependencies, whereas lower layers are often dominated by local token-level processing (Tenney et al., 2019a;b; Jawahar et al., 2019). We further support this design choice with a layer-selection ablation in Tab. 10, which shows that later layers provide more reliable signals for recovering the underlying dependency structure.

**Separability of edges and non-edges.** We first evaluate whether the attention-based score $s_{ij}$ can distinguish ground-truth edges in the MRF from non-edges. We compare the empirical distributions of $s_{ij}$ for the two groups and report (i) the area under the ROC curve (AUC) for edge classification using $s_{ij}$ and (ii) the ratio of their mean scores. Across multiple runs, $s_{ij}$ exhibits clear separation between edges and non-edges, achieving AUC of 0.928 and edge-to-non-edge score ratio of 2.204 (see Tab. 1). These results suggest that attention provides a reliable signal for recovering the underlying MRF structure.

**Degree estimation from attention sums.** While the interaction score $s_{ij}$ provides strong separability, choosing an exact threshold $\tau_t$ to recover the exact edge set remains challenging. However, our proposed algorithm (Sec. 4) relies primarily on *sorting* nodes by degree, rather than on estimating the exact edge set. Consequently, we focus on validating whether simple attention aggregates can serve as a robust proxy for the true node degree in the ground-truth MRF. We examine whether the sum of edge scores

$$\tilde{d}_i := \sum_{j \in V_t \setminus \{i\}} s_{ij}$$

aligns with the true degree $d_i$ of node $i$. We introduce an intuitive measure, the *Order Violation Rate* (OVR), defined as the fraction of pairs whose strict order under the true degrees is reversed by the proxy scores. Formally,

$$\text{OVR}(t) = \frac{1}{\binom{|V_t|}{2}} \sum_{d_i < d_j} \mathbb{1}[\tilde{d}_i > \tilde{d}_j],$$

Lower OVR indicates better agreement with the true ordering. We find that the ordering is highly consistent (mean OVR $= 0.04$; Tab. 1), suggesting that the proxy $\tilde{d}_i$ provides a reliable estimate for node degree, and thus for the relative structural importance of the corresponding variable.

## 4. Dependency-Aware Parallel Decoding

In Sec. 3, we established that the interactions among masked tokens can be modeled as a Markov Random Field (MRF), denoted as $G_t$. We now leverage this structural understanding to guide parallel decoding. At each decoding step, our goal is to select a subset of masked tokens that are approximately conditionally independent given the context, thereby reducing joint–marginal mismatch and enabling parallel generation that better matches the true joint distribution. By maximizing the size of such a valid subset that satisfies these dependency constraints, we aim to effectively reduce the total number of decoding steps (NFE), the primary computational bottleneck identified in Sec. 2.2, without compromising generation quality.

### 4.1. Independent Set Selection via Structural Relaxation

To determine the set of tokens to unmask at step $t$, we select an *independent set* $S \subseteq V_t$, defined as a subset of nodes with no direct edges between them. Theoretically, a sufficient condition for full joint independence in an MRF is that the nodes are pairwise disconnected (i.e., no path exists between any pair in $S$) rather than merely non-adjacent (Koller & Friedman, 2009). However, we adopt pairwise non-adjacency as a sufficient condition for practical decoding. Enforcing strict separation incurs substantial verification cost and yields prohibitively small batches, limiting parallel speedup. Furthermore, in natural language, dependencies are predominantly captured by direct interactions; instances of joint dependence arising solely from indirect paths are negligible. Thus, this relaxation effectively mitigates the risk of severe incoherence while maximizing efficiency.

### 4.2. Parallel Decoding as Dynamic Graph Coloring

Given the MRF constructed above, parallel decoding reduces to the problem of selecting independent sets in the graph. In dLLM decoding, our objective is to minimize the number of decoding steps, which is equivalent to covering all masked positions with as few independent sets as possible. This directly maps the problem to *graph coloring*: in a graph, a proper coloring partitions nodes into disjoint independent sets (color classes), and the minimum number of colors required (the chromatic number) corresponds to the minimum number of parallel decoding steps.

While classical graph coloring is defined on a static graph, our setting differs in that the MRF $G_t = (V_t, E_t)$ is dynamic. After each decoding step, the set of masked nodes $V_t$ shrinks and the edge structure $E_t$ is perturbed as new context becomes available. Nevertheless, the core combinatorial intuition remains applicable. At each step, selecting an independent set can be viewed as a coloring of the current graph, with the goal of minimizing the total number of decoding steps over time. This motivates us to adopt graph-coloring-inspired strategies for parallel decoding.

### 4.3. Implementation via a Coloring Strategy

We now operationalize the graph-coloring perspective by designing a sequential independent-set selection procedure. A natural greedy objective is to maximize the number of nodes decoded at each step, i.e., to select a maximum independent set of the current graph. However, our goal is to minimize the total number of steps, which corresponds to covering the node set with as few independent sets (color classes) as possible. These objectives need not coincide: heuristics that favor large independent sets often prioritize low-degree nodes, whereas reducing the number of future color classes benefits from resolving high-degree nodes early.

Accordingly, we adopt the *Welsh–Powell algorithm* (Welsh & Powell, 1967), a degree-prioritized greedy coloring strategy. At each step, we order the remaining nodes by non-increasing degree and then scan this list, adding a node whenever it is non-adjacent to the nodes already selected. This procedure constructs a maximal (not necessarily maximum) independent set to be decoded in parallel. Although the resulting set may be smaller than a maximum independent set in the current step, prioritizing high-degree nodes tends to simplify the residual graph, making it easier to cover with fewer independent sets in subsequent steps and ultimately reducing the total number of decoding steps.

**Method.** We instantiate this algorithm for dLLM decoding as **Dependency-Aware Parallel Decoding (DAPD)** to construct our parallel batches. At each decoding step $t$, we approximate the MRF $G_t = (V_t, E_t)$ using the attention-derived edge scores $s_{ij}$. Motivated by the layer-selection analysis in Sec. 3.2, we compute the edge scores in our main experiments (Sec. 5) by averaging attention weights over all heads and the final 30% of Transformer layers. We estimate each node degree with $\tilde{d}_i$ and define edges by thresholding $s_{ij}$ with a conservative, low threshold $\tau_t$. This threshold declares a non-edge only when the interaction is confidently negligible, thereby prioritizing conflict avoidance in independent-set selection over edge-set precision.

Given this proxy graph, we construct an independent set $S$ using the Welsh-Powell-motivated strategy:

1. **Ordering:** We sort all masked nodes in descending order of their proxy degree $\tilde{d}_i$.

2. **Greedy Selection:** We iterate through the list, adding a node to $S$ if and only if it is non-adjacent to all nodes currently in $S$.

3. **Update:** All tokens in $S$ are unmasked simultaneously. The graph is then recomputed for the next step.

From a dLLM perspective, this strategy prioritizes positions that strongly interact with many others. Resolving these "hub" nodes early effectively breaks the major dependency chains, sparsifying the graph for subsequent steps and enabling larger parallel batches later in the process.

This graph construction adds only negligible overhead relative to the main bottleneck in dLLM decoding: model forward passes. DAPD reuses the self-attention weights already computed during the forward pass, so dependency estimation and independent-set selection require no additional model evaluations. As a result, it reduces the dominant inference cost by decreasing the number of decoding steps, while adding only minor graph-processing burden.

**Practical Implementation.** In our final implementation, we refine the sorting metric to a confidence-weighted degree $\tilde{d}_i \cdot \mathrm{conf}_i$, where $\mathrm{conf}_i$ denotes the marginal confidence at position $i$, the maximum predicted token probability. Since $\mathrm{conf}_i$ reflects the probability that the most likely token at position $i$ will be correctly generated, multiplying $\tilde{d}_i$ by $\mathrm{conf}_i$ can be interpreted as measuring the *expected effective degree* of node $i$. That is, while $\tilde{d}_i$ captures the structural influence of a position in the dependency graph, $\tilde{d}_i \cdot \mathrm{conf}_i$ quantifies how much of this influence is expected to be realized given the model's current uncertainty. This criterion prioritizes nodes that are both structurally important and reliably predictable, aligning the ordering metric with the goal of reducing downstream conflicts in iterative decoding.

As decoding progresses and strong dependencies are resolved, the MRF becomes sparse. When the remaining mask ratio falls below 50%, we switch on a confidence-threshold rule, unmasking all positions with confidence $> 0.9$ (Wu et al., 2026). In graph terms, this corresponds to a regime where most remaining nodes have near-zero degree, i.e., are conditionally independent given the resolved context. In this disconnected regime, confidence-based sampling effectively serves as an aggressive independent-set approximation, enabling efficient completion. We empirically verify this sparsification behavior in Sec. 6.

*Remark* 4.1. In App. A, we further evaluate another confidence-based acceleration variant. When positions have confidence 1.0, their predicted tokens are effectively deterministic. Indeed, if a marginal distribution assigns probability one to a token, then any compatible joint distribution must also assign probability one to that token at the corresponding position. Thus, the variant preemptively unmasks all confidence-1.0 positions throughout decoding, yielding a more aggressive accuracy–steps trade-off without introducing joint–marginal mismatch on those positions.

## 5. Experiments

**Setup.** We evaluate DAPD on two representative open-source dLLMs: LLaDA-8B-Instruct (Nie et al., 2025) and Dream-7B-Instruct (Ye et al., 2025). We compare against training-free parallel decoding baselines, including EB-Sampler (Ben-Hamu et al., 2025), KLASS (Kim et al., 2025b), and Fast-dLLM (Wu et al., 2026). For all baselines, we adopt a top-1 confidence-based unmasking strategy and use the best hyperparameters reported in the original papers.

Following prior works (Ben-Hamu et al., 2025; Kim et al., 2025b; Wu et al., 2026), we evaluate DAPD on math reasoning (GSM8K (Cobbe et al., 2021), Math500 (Hendrycks et al., 2021)) and code generation (MBPP (Austin et al., 2021b), HumanEval (Chen et al., 2021)). We further include IFEval (Zhou et al., 2023) to assess instruction-following capability. All evaluations use the `lm-eval` framework

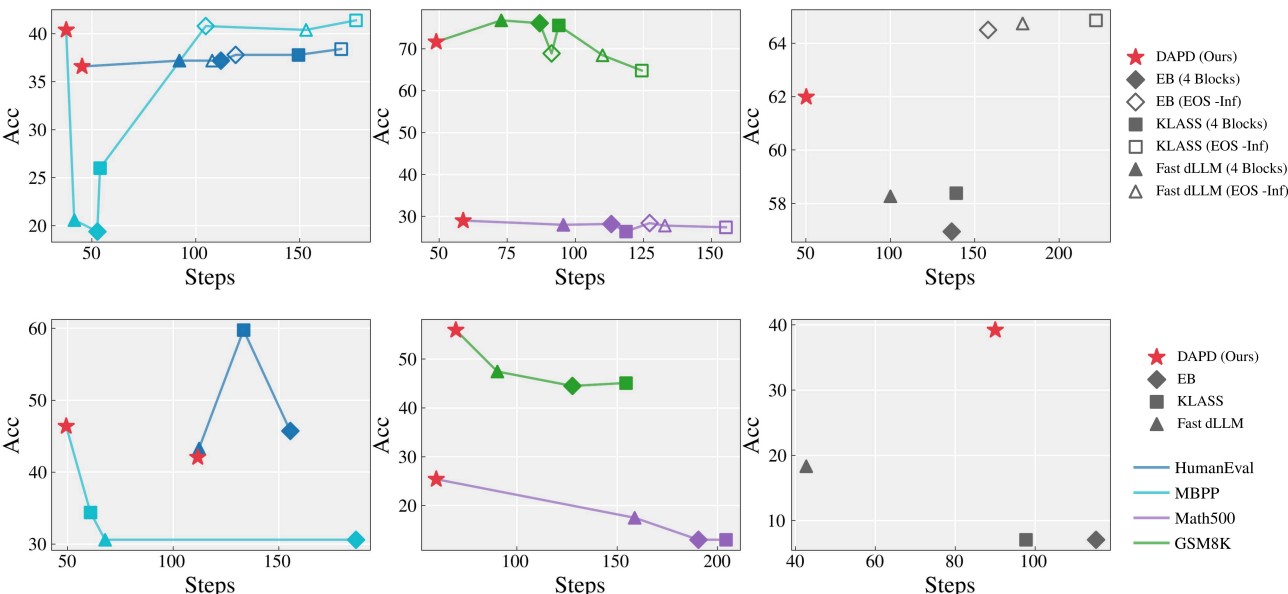

*Figure 3.* Accuracy–Steps trade-off of decoding strategies on two dLLMs. **Top**: LLaDA. **Bottom**: Dream. **Left**: Code tasks (HumanEval, MBPP). **Middle**: Math tasks (GSM8K, Math500). **Right**: Instruction-following task (IFEval). Markers denote decoding strategies. For LLaDA, baselines are shown with block decoding and EOS suppression settings ("4 Blocks" and "EOS-Inf"), while DAPD is evaluated in the single-block regime by default. For Dream, all methods are evaluated in the single-block regime. Colored lines indicate different tasks. The exact numerical values corresponding to this plot are provided in Tab. 3.

([Gao et al.](), 2024) with a maximum generation length of 256 tokens. We also report ParallelBench ([Kang et al.](), 2026), which stress-tests parallel decoding under strong inter-token dependencies and joint–marginal mismatch. Full experimental details and configurations are provided in App. A.

**Results.** We begin by evaluating training-free baselines on LLaDA under 1-block decoding; however, their accuracy collapses in this regime, making direct comparison impractical (see Tab. 5). We find that this degradation is primarily driven by premature termination due to EOS overflow ([Kim et al.](), 2026). To ensure a fair evaluation, we follow the protocols established in the official LLaDA implementation ([Nie et al.](), 2025), applying block-wise decoding or EOS suppression to the baselines. In contrast, our method is evaluated using a single block by default, highlighting its robustness under single-block parallel decoding.

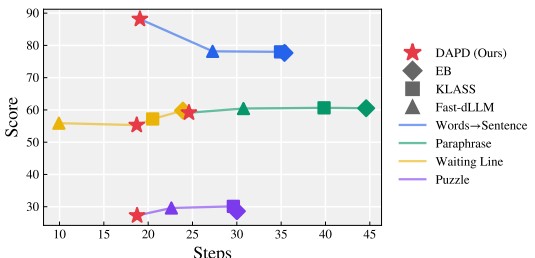

*Figure 4.* Score–steps trade-off on ParallelBench using LLaDA. Exact numerical values are provided in Tab. 4.

Fig. 3 demonstrates the performance of various decoding strategies across the accuracy–steps trade-off. Notably, our method consistently occupies the upper-left region. In particular, on MBPP and IFEval, DAPD attains significantly higher accuracy than block-wise baselines. Although single-block decoding with EOS suppression achieves comparable accuracy, it requires significantly more decoding steps. On the remaining tasks, DAPD achieves comparable accuracy while providing substantial speedups over other methods. We further provide end-to-end throughput measurements in App. A, showing that the reduction in decoding steps translates into improved tokens-per-second (TPS) despite the additional graph-processing overhead. Additionally, as shown in Fig. 4, our method achieves a more favorable Score-Steps trade-off than the baselines on most ParallelBench tasks. This indicates that our approach more effectively identifies and updates low-dependency token sets in parallel, even under complex dependency structures.

Unlike LLaDA, Dream does not support block-wise decoding or EOS suppression in its official implementation. Moreover, we observe that Dream does not exhibit severe EOS overflow at a generation length of 256 tokens. Accordingly, we evaluate all methods under a unified single-block decoding setup. As shown in Fig. 3, our method maintains a superior accuracy–steps trade-off relative to the baselines. This trend aligns with our findings on LLaDA, suggesting that our approach generalizes across diverse dLLMs.

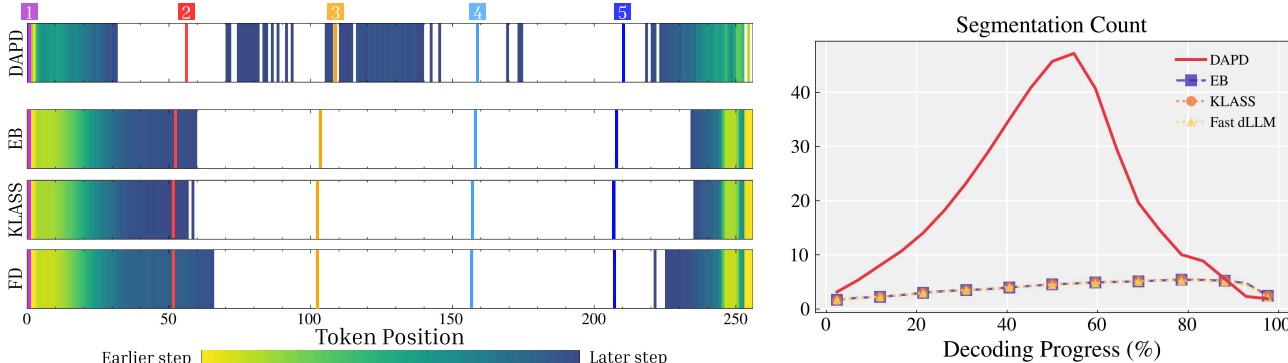

*Figure 5.* **Left:** Distribution of tokens unmasked during the initial 40% of decoding. Progress is normalized by total steps per sample and method. Heatmaps show the average unmasking trajectory, where lighter colors indicate earlier unmasking and white regions denote tokens remaining masked. Vertical color bars (labeled above) indicate average starting positions for the five questions; FD denotes Fast-dLLM. **Right:** Average number of isolated segments per decoding step, characterizing sequence fragmentation. Notably, DAPD exhibits a distinct unmasking pattern, whereas baselines show highly similar behaviors.

## 6. Analysis of Decoding Behavior

In this section, we examine the token unmasking dynamics of different decoding strategies. Ideally, an optimal parallel decoding should identify and exploit conditionally independent subsets of tokens under the current context, enabling them to be decoded simultaneously and thus maximizing computational throughput. However, such independence is difficult to isolate in standard benchmarks, which typically evaluate generation quality on a single globally coherent answer, thereby inducing strong semantic and statistical coupling across tokens.

**Setup.** With this in mind, we turn to a setting where conditional independence naturally arises: prompts that bundle multiple independent queries into a single generation. Specifically, we draw samples from TriviaQA (Joshi et al., 2017), a fact-oriented benchmark characterized by concise and unambiguous answers. We aggregate five randomly sampled questions into a single prompt, resulting in a total of 500 evaluation samples (2,500 individual questions).

All methods are evaluated on LLaDA with a maximum generation budget of 256 tokens. To isolate the effects of the decoding strategies, we disable block-wise decoding. We do not observe EOS overflow here, as the input prompts are sufficiently short. Details regarding dataset construction and the broader experimental setup are provided in App. C.

**Results.** Fig. 5 and Tab. 2 summarize the qualitative and quantitative behaviors induced by different decoding strategies. Fig. 5 (left) shows that DAPD exhibits a qualitatively distinct decoding pattern compared to baselines that rely on marginal confidence, while Fig. 5 (right) reports the corresponding evolution of the *segment count*, defined as the number of disjoint contiguous segments of unmasked tokens in the generation region at each step. DAPD follows a characteristic trajectory in which the segment count in-

*Table 2.* Accuracy and average decoding steps, with relative speedup measured against step-by-step LLaDA decoding strategy. "Original" denotes confidence-based token-by-token decoding.

| Method | Acc. ↑ | Steps (Speedup) ↓ |
|---|---|---|
| Original | 52.64 | 256.0 (1.00×) |
| Fast-dLLM | 52.12 | 124.4 (2.06×) |
| KLASS | 52.2 | 177.4 (1.44×) |
| EB-Sampler | 51.2 | 131.3 (1.95×) |
| DAPD (Ours) | 52.08 | 66.2 (3.87×) |

creases toward the midpoint of generation and then drops sharply as segments merge. In contrast, baseline methods maintain only a small number of segments throughout decoding. These structural differences directly translate into efficiency gains. As shown in Tab. 2, DAPD matches the accuracy of LLaDA with step-by-step decoding while significantly reducing the number of generation steps, achieving more than $1.8\times$ speedup over competing strategies and a $3.8\times$ speedup compared to standard step-by-step decoding by solving independent questions concurrently.

**Sequential bottlenecks in baselines.** Fig. 5 indicates that baseline methods exhibit highly localized decoding trajectories. Despite the independence of the queries, these approaches predominantly unmask adjacent tokens, progressively focusing on contiguous regions and resolving queries sequentially. As a result, the segment count remains low, reflecting an autoregressive-like behavior constrained by spatial locality. This highlights a key limitation of relying solely on marginal confidence as a decoding critic: marginal scores are heavily influenced by local context and fail to promote concurrent generation across distant regions. Consequently, baseline methods are unable to fully exploit the throughput benefits of parallel decoding or the any-order flexibility that dLLMs are designed to support.

**Spatially dispersed generation in DAPD.** In contrast, DAPD encourages a spatially dispersed, largely order-agnostic unmasking behavior across the five independent queries from early decoding stages. By explicitly accounting for inter-token dependencies, DAPD avoids committing to a single contiguous region and instead distributes unmasking decisions across distant positions. This dispersed behavior leads to the formation of multiple disjoint segments early in generation, as reflected by the rise in segment count. Importantly, early spatially dispersed unmasking provides partial context across all queries, which subsequently reduces uncertainty when filling the remaining masked spans and enables rapid consolidation in later steps. Taken together, this behavior more faithfully realizes the core promise of dLLMs: leveraging bidirectional context to support truly any-order generation, rather than collapsing into an autoregressive-like trajectory.

## 7. Conclusion

We propose Dependency-Aware Parallel Decoding (DAPD), a training-free decoding method that directly targets joint–marginal mismatch in dLLM parallel decoding. Rather than relying only on marginal confidence, DAPD uses self-attention as a model-internal signal for estimating cross-position interactions among masked tokens. At each decoding step, DAPD converts these attention signals into an attention-induced dependency graph and selects an independent set of weakly coupled positions for parallel unmasking. Across LLaDA and Dream, DAPD improves the accuracy–steps trade-off over strong training-free baselines and shifts decoding from locally clustered behavior to globally dispersed, order-flexible generation.

More broadly, our work suggests that self-attention can be repurposed as a lightweight dependency proxy for decoding, complementing marginal confidence with cross-position interaction information. This perspective reduces dLLM parallel decoding to a dynamic graph-coloring problem, where generation is guided not only by token-wise reliability but also by the evolving structure of inter-token dependencies. We believe this graph-based view provides a useful foundation for future parallel decoding algorithms that move beyond purely marginal token selection.

## Acknowledgements

This work was supported in part by Institute of Information & communications Technology Planning & Evaluation (IITP) grant funded by the Korea government (MSIT) (No. RS-2024-00457882, AI Research Hub Project), and the National Research Foundation of Korea (NRF) grant funded by the Korea government (MSIT) (No. RS-2024-00410005, RS-2025-23525649).

## Impact Statement

This paper presents work whose goal is to improve the efficiency of diffusion-based language model inference. By reducing the number of decoding steps without additional training or auxiliary models, the proposed method may make dLLM inference more practical and computationally efficient. We do not anticipate direct societal impacts beyond those generally associated with language generation models.

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

*Table 3.* Numerical results for the LLaDA and Dream experiments, including the settings visualized in Fig. 3. For LLaDA, the baseline results (Fast-dLLM, EB-Sampler, and KLASS) are reported under four-block decoding, since their single-block variants suffer severe performance degradation due to EOS overflow. For Dream, all methods are evaluated under a unified single-block decoding setup. DAPD-STAGED denotes the main variant that applies dependency-graph selection and, after the remaining mask ratio falls below 50%, admits positions with confidence $> 0.9$ into the independent-set selection procedure. DAPD-DIRECT denotes the optional acceleration variant that, at each decoding step, first commits positions with confidence $= 1.0$ and then applies dependency-aware independent-set selection to the remaining masked positions. EOS-suppressed single-block baseline results are reported separately in Tab. 5.

| Method | HumanEval | | MBPP | | GSM8K | | Math500 | | IFEval | |
|---|---|---|---|---|---|---|---|---|---|---|
| | Acc. | Steps | Acc. | Steps | Acc. | Steps | Acc. | Steps | Acc. | Steps |
| *LLaDA* | | | | | | | | | | |
| Fast-dLLM | 37.2 | 92.1 | 20.6 | 41.5 | 76.8 | 72.8 | 28.0 | 95.6 | 58.3 | 100.0 |
| EB-Sampler | 37.2 | 110.4 | 19.4 | 52.6 | 76.1 | 86.9 | 28.2 | 113.2 | 57.0 | 136.3 |
| KLASS | 37.8 | 149.4 | 26.0 | 53.9 | 75.6 | 93.9 | 26.4 | 118.6 | 58.4 | 139.0 |
| DAPD-Staged | 36.6 | 45.2 | 40.4 | 37.6 | 71.7 | 48.9 | 29.0 | 58.8 | 62.0 | 50.1 |
| DAPD-Direct | 37.2 | 22.7 | 37.2 | 20.8 | 70.9 | 32.5 | 27.2 | 45.9 | 57.3 | 42.9 |
| *Dream* | | | | | | | | | | |
| Fast-dLLM | 43.3 | 112.2 | 30.6 | 67.7 | 47.7 | 90.2 | 17.5 | 158.7 | 18.2 | 53.2 |
| EB-Sampler | 45.7 | 155.4 | 30.6 | 186.5 | 44.5 | 127.6 | 13.0 | 190.5 | 7.1 | 115.2 |
| KLASS | 59.8 | 133.3 | 34.4 | 60.8 | 45.1 | 154.4 | 13.0 | 204.2 | 7.1 | 132.1 |
| DAPD-Staged | 42.1 | 111.6 | 46.4 | 49.3 | 56.0 | 69.5 | 25.4 | 59.7 | 39.2 | 90.0 |
| DAPD-Direct | 43.3 | 117.8 | 48.2 | 26.9 | 59.4 | 55.6 | 26.4 | 56.0 | 38.7 | 18.6 |

*Table 4.* Numerical results for ParallelBench, including the settings visualized in Fig. 4.

| Method | Words→Sentence | | Paraphrase | | Waiting Line | | Puzzle | |
|---|---|---|---|---|---|---|---|---|
| | Score | Steps | Score | Steps | Score | Steps | Score | Steps |
| Fast-dLLM | 78.2 | 27.3 | 60.4 | 30.8 | 55.9 | 9.9 | 29.6 | 22.6 |
| EB-Sampler | 77.7 | 35.4 | 60.5 | 44.6 | 59.8 | 23.9 | 28.6 | 30.0 |
| KLASS | 78.0 | 34.9 | 60.7 | 39.8 | 57.2 | 20.5 | 30.1 | 29.6 |
| DAPD-Staged | 88.2 | 19.1 | 59.2 | 24.6 | 55.3 | 18.7 | 27.3 | 18.8 |
| DAPD-Direct | 86.5 | 17.9 | 59.4 | 18.9 | 54.8 | 10.8 | 23.4 | 20.1 |

## A. Details on Main Experiments

**Basic setup.** All experiments are conducted on NVIDIA L40S GPUs. For consistent and reproducible evaluation, we disable stochastic decoding parameters such as temperature. We use the `lm-eval` framework for all evaluations, following its default setting for the number of few-shot examples. These settings are applied consistently across both LLaDA and Dream. For LLaDA, we additionally report results under both single-block and four-block decoding for the training-free baselines. As shown in Tab. 5, baselines suffer severe performance degradation in the single-block decoding due to premature termination from *EOS overflow* (Kim et al., 2026), whereas four-block decoding substantially mitigates this issue.

**Results.** The main text presents the accuracy–steps and score–steps trade-offs primarily through plots (Figs. 3 and 4). For completeness, we report the corresponding numerical values in Tab. 3 for the standard benchmarks on LLaDA and Dream, and in Tab. 4 for ParallelBench. For DAPD, we separate the default setting from an acceleration-oriented variant. DAPD-STAGED, used throughout the main text and the other appendices, applies dependency-aware selection throughout decoding and adds the confidence-threshold shortcut (confidence $> 0.9$) only after the remaining mask ratio falls below 50%. DAPD-DIRECT, discussed in Remark 4.1, first commits confidence-$1.0$ positions at each step before running dependency-aware selection on the remaining masked positions. Overall, DAPD-STAGED gives the most stable accuracy–steps trade-off while substantially reducing decoding steps. DAPD-DIRECT is more aggressive, usually reducing steps further with modest accuracy changes and occasional score gains. We therefore treat DAPD-DIRECT as a latency-oriented optional variant and keep DAPD-STAGED as the default.

*Table 5.* Accuracy (%) and average decoding steps under single-block, EOS-suppressed single-block, and four-block decoding. Without EOS suppression, the single-block baselines suffer severe performance degradation due to *EOS overflow* (Kim et al., 2026). EOS-Inf mitigates this collapse, but typically requires substantially more decoding steps. The four-block setting is used for the main comparison.

| Method | Setting | HumanEval | | MBPP | | GSM8K | | Math500 | | IFEval | |
|---|---|---|---|---|---|---|---|---|---|---|---|
| | | Acc. | Steps | Acc. | Steps | Acc. | Steps | Acc. | Steps | Acc. | Steps |
| KLASS | 1 block | 11.0 | 97.3 | 19.8 | 43.3 | 26.3 | 72.6 | 3.0 | 83.4 | 40.1 | 96.5 |
| | 1 block + EOS-Inf | 38.4 | 170.0 | 41.4 | 177.1 | 64.8 | 124.4 | 27.4 | 155.3 | 64.9 | 221.8 |
| | 4 blocks | 37.8 | 149.4 | 26.0 | 53.9 | 75.6 | 93.9 | 26.4 | 118.6 | 58.4 | 139.0 |
| Fast-dLLM | 1 block | 10.4 | 40.7 | 9.6 | 34.2 | 7.5 | 89.4 | 1.8 | 76.4 | 41.7 | 31.7 |
| | 1 block + EOS-Inf | 37.2 | 107.9 | 40.4 | 153.1 | 68.5 | 110.0 | 27.8 | 132.9 | 64.7 | 178.4 |
| | 4 blocks | 37.2 | 92.1 | 20.6 | 41.5 | 76.8 | 72.8 | 28.0 | 95.6 | 58.3 | 100.0 |
| EB-Sampler | 1 block | 13.4 | 85.9 | 8.0 | 61.1 | 6.6 | 143.4 | 2.0 | 136.3 | 30.7 | 108.3 |
| | 1 block + EOS-Inf | 37.8 | 119.2 | 40.8 | 104.8 | 68.9 | 91.3 | 28.4 | 127.3 | 64.5 | 157.9 |
| | 4 blocks | 37.2 | 112.1 | 19.4 | 52.6 | 76.1 | 86.9 | 28.2 | 113.2 | 57.0 | 136.3 |

**Hyperparameter details.** For commonly used benchmarks (HumanEval, MBPP, Math500, GSM8K), we adopt the best hyperparameter settings for baseline methods (EB-Sampler, KLASS, Fast-dLLM) as reported in their original papers or official code implementations. We refer readers to the respective works for detailed configurations. As there are no established hyperparameter references for the IFEval task, we perform hyperparameter sweeps for each baseline method. For EB-Sampler, we vary the parameter $\gamma$ over $\{0.05, 0.1, 0.15, 0.2\}$ for LLaDA and Dream. For Fast-dLLM and KLASS, we sweep the confidence threshold over $\{0.8, 0.85, 0.9, 0.95\}$. For KLASS, we sweep the KL threshold over $\{0.001, 0.005\}$ for both LLaDA and Dream. After sweeping, we select the configuration that achieves the most favorable trade-off between accuracy and steps, and report those results in the main section.

For DAPD, we use a hyperparameter $\tau_t$ to determine whether a token is selected for sampling based on its attention-based edge score. We adopt a simple linear schedule in which $\tau_t$ increases over the decoding process. We report two DAPD variants: DAPD-Staged, which uses the staged confidence-threshold rule described in the main text, and DAPD-Direct, which applies direct commits for confidence-1.0 positions throughout decoding. For DAPD-Staged, LLaDA uses $[0.01, 0.05]$

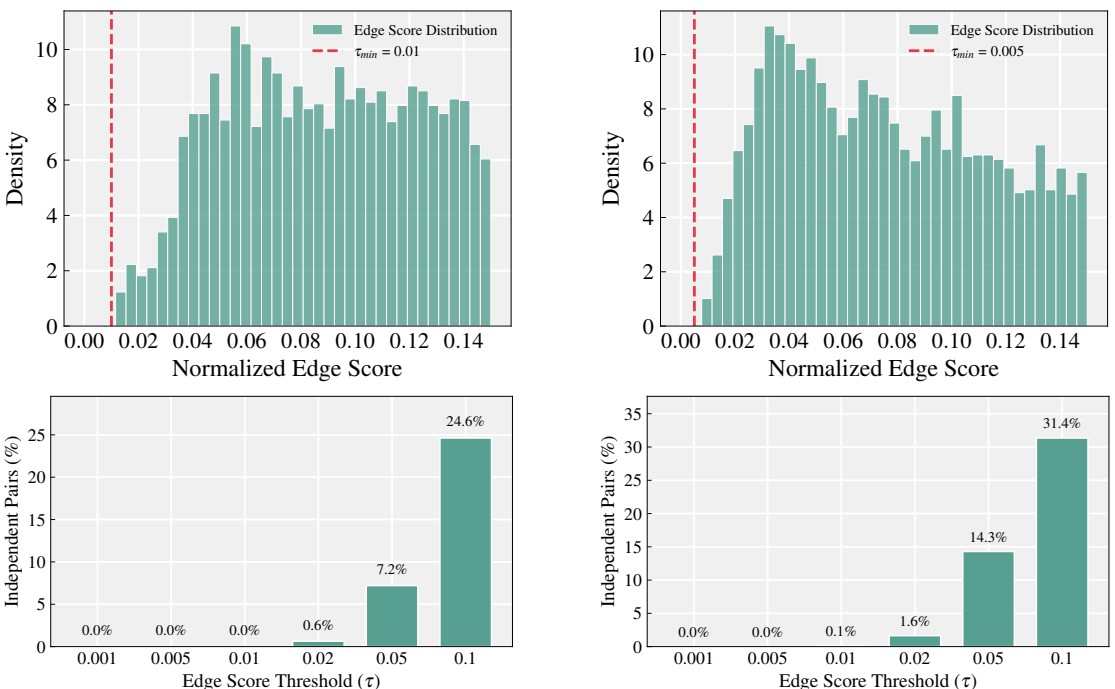

*Figure 6.* **Analysis of $\tau_{\min}$.** Left: LLaDA. Right: Dream.

for math tasks (GSM8K and Math500) and $[0.01, 0.15]$ for the remaining tasks. For Dream, we use a linear $\tau_t$ schedule over $[0.005, 0.05]$ across all tasks; for IFEval, we use $[0.1, 0.5]$. For DAPD-Direct, LLaDA uses $[0.005, 0.05]$ for GSM8K and Math500, $[0.01, 0.05]$ for HumanEval and IFEval, and $[0.01, 0.02]$ for MBPP. Dream uses $[0.005, 0.01]$ for HumanEval, MBPP, GSM8K, and Math500, and $[0.1, 0.5]$ for IFEval.

To motivate our choice of $\tau_{\min}$, we empirically characterize the distribution of *normalized mask-to-mask edge scores* under the Sec. 6 analysis setup. Specifically, for both LLaDA and Dream with generation length 256, we run decoding step-by-step and, at each step, collect pairwise edge scores among the remaining masked positions. As shown in Fig. 6, we observe that the chosen values, $\tau_{\min} = 0.01$ for LLaDA and $\tau_{\min} = 0.005$ for Dream, fall in the near-zero tail of the observed edge score distribution, so the early-stage decoding admits only very weak interactions between concurrently updated positions. In practice, decreasing $\tau_{\min}$ further has little effect on the available set of low-dependency pairs, as the fraction of pairs below such small thresholds remains close to zero in our measurements.

**Details on ParallelBench.** We additionally run ParallelBench (Kang et al., 2026), a benchmark designed to assess parallel decoding capability, using its native evaluation protocol. ParallelBench comprises multiple task types with task-specific generation lengths and instance sizes. Across tasks, $n$ denotes the task-specific problem size (instance size), e.g., the number of entities in *Waiting line*, the grid/order in *Latin square* (and related puzzle settings), or a difficulty parameter in *Words to Sentence*. We evaluate the following task types and subtasks: **Waiting Line** ($n = 15$) (10 subtasks; generation length = 128; we use the n=15 variant rather than the default waiting line since the latter uses generation length = 32, which is less suitable for evaluating parallel decoding): copy, insert (index), insert (random), remove (index), remove (random), replace (index), replace (random), reverse, shuffle, sort. **Puzzle** (2 subtasks; generation length = 64): Latin square ($n = 4$), Sudoku ($n = 4$, 12 clues). **Paraphrase and Summarize** (2 subtasks; generation length = 128): ChatGPT paraphrase, SAMSum summarization. **Words to Sentence** (6 subtasks; generation length = 64): easy, easy ($n = 1$), easy ($n = 3$), easy ($n = 4$), easy ($n = 5$), easy ($n = 6$).

For DAPD on ParallelBench, we use a single shared linear schedule across all task types: $\tau_t \in [0.01, 0.2]$ for DAPD-Staged and $\tau_t \in [0.01, 0.05]$ for DAPD-Direct. For training-free baselines, we use their default hyperparameters under the LLaDA setup: Fast-dLLM uses confidence threshold 0.9; EB-Sampler uses $\gamma = 0.1$; KLASS uses confidence threshold 0.9 and KL threshold 0.01. All other settings follow the official ParallelBench protocol.

*Table 6.* End-to-end throughput comparison on HumanEval. Accuracy, average decoding steps, and tokens per second (TPS) are reported. "Original" denotes confidence-based token-by-token decoding.

| Method | Acc. ↑ | Steps ↓ | TPS ↑ |
|---|---|---|---|
| DAPD | 36.6 | 45.2 | 106.0 |
| Fast-dLLM | 37.2 | 92.1 | 51.4 |
| EB-Sampler | 37.2 | 110.4 | 39.2 |
| KLASS | 37.8 | 149.4 | 25.6 |
| Original | 39.0 | 256.0 | 20.4 |

**End-to-End Throughput.** While the main experiments focus on the accuracy–steps trade-off, fewer decoding steps do not necessarily imply better wall-clock efficiency. DAPD adds graph-processing operations, including thresholding, degree estimation, and greedy independent-set selection, but these are lightweight compared to repeated model forward passes. Moreover, DAPD reuses the self-attention weights already computed during the forward pass, introducing no additional model evaluations. To verify practical speedup, we measure end-to-end throughput in tokens per second (TPS) on HumanEval. As shown in Tab. 6, DAPD achieves the highest TPS among all compared methods. TPS is also strongly aligned with the inverse of the number of decoding steps, suggesting that DAPD's graph-processing overhead is negligible relative to model forward passes. Thus, dependency-aware selection improves practical inference efficiency, rather than merely reducing the nominal number of decoding steps.

*Table 7.* Longer-generation evaluation of DAPD-Staged. Accuracy, average decoding steps, and tokens per second (TPS) are reported for different maximum generation lengths.

| Task | Length | Acc. ↑ | Steps ↓ | TPS ↑ |
|------|--------|--------|---------|-------|
| HumanEval | 256 | 36.6 | 45.2 | 106.0 |
|  | 512 | 32.9 | 55.9 | 106.6 |
|  | 1024 | 31.7 | 67.5 | 93.6 |
| GSM8K | 256 | 70.7 | 48.2 | 33.9 |
|  | 512 | 72.8 | 68.4 | 39.0 |
|  | 1024 | 70.7 | 79.9 | 45.8 |

**Longer Generation Lengths.** We further evaluate whether the practical efficiency of DAPD persists under longer generation lengths. Since dependency graph construction is repeated at every decoding step, one potential concern is that its overhead may become more pronounced as the number of masked positions increases. To examine this, we measure accuracy, the average number of decoding steps, and TPS under maximum generation lengths $L \in \{256, 512, 1024\}$. As shown in Tab. 7, DAPD maintains strong end-to-end throughput even as the generation length increases. The number of decoding steps grows moderately with length, while TPS remains largely stable across longer generations. This indicates that the lightweight graph-processing overhead does not dominate the inference cost. These results suggest that DAPD scales favorably beyond the default 256-token setting used in the main experiments.

*Table 8.* Block-wise decoding evaluation on HumanEval. Accuracy and tokens per second (TPS) are reported under different numbers of generation blocks.

| Method | Blocks | Acc. ↑ | TPS ↑ |
|--------|--------|--------|-------|
| DAPD | 1 | 36.6 | 106.0 |
|  | 4 | 37.8 | 57.1 |
|  | 8 | 39.6 | 43.6 |
|  | 16 | 41.5 | 34.6 |
| Fast-dLLM | 4 | 37.2 | 51.4 |
| EB-Sampler | 4 | 37.2 | 39.2 |
| KLASS | 4 | 37.8 | 25.6 |

**Block-Wise Decoding.** Single-block decoding is the most natural setting for DAPD, since it allows the dependency graph to be constructed over all remaining masked positions and therefore maximally exploits global parallelism. Nevertheless, following the block-wise generation setup commonly used in dLLMs (Arriola et al., 2025), we also evaluate DAPD under multi-block decoding. As shown in Tab. 8, DAPD remains compatible with block-wise generation and achieves competitive or better accuracy than the baselines when using 4, 8, or 16 blocks. Increasing the number of blocks tends to improve accuracy, as the left-to-right block structure restricts the generation problem to shorter local regions. However, this comes at the cost of reduced throughput: once generation is divided into sequential blocks, DAPD can only select independent sets within the current block, rather than across the full sequence. Thus, even though each local dependency graph becomes smaller, the method loses part of its global parallelism across distant independent positions. These results show that DAPD is not restricted to the single-block regime, while its single-block configuration better reflects the globally parallel decoding capability of dLLMs.

*Table 9.* Edge detection and degree estimation metrics across decoding steps, reported as mean $\pm$ standard deviation over sampling paths.

| Steps | Edge Detection | | Degree Estimation |
|---|---|---|---|
| | AUC $\uparrow$ | Ratio (Edge/Non-edge) $\uparrow$ | OVR $\downarrow$ |
| 1 | $0.923 \pm 0.00$ | $2.19 \pm 0.00$ | $0.00 \pm 0.00$ |
| 2 | $0.923 \pm 0.02$ | $2.21 \pm 0.12$ | $0.04 \pm 0.01$ |
| 3 | $0.919 \pm 0.04$ | $2.18 \pm 0.23$ | $0.05 \pm 0.01$ |
| 4 | $0.916 \pm 0.04$ | $2.17 \pm 0.21$ | $0.08 \pm 0.02$ |
| 5 | $0.937 \pm 0.05$ | $2.23 \pm 0.24$ | $0.05 \pm 0.01$ |
| 6 | $0.937 \pm 0.10$ | $2.17 \pm 0.37$ | $0.04 \pm 0.02$ |
| 7 | $0.944 \pm 0.12$ | $2.28 \pm 0.45$ | $0.02 \pm 0.04$ |
| 8 | – | – | – |
| 9 | – | – | – |

## B. Details on Toy Experiments

Here, we provide additional details regarding the graph experiments discussed in Sec. 3.2. We employ the DiT (Peebles & Xie, 2023) architecture within the RADD (Ou et al., 2025) framework, configured with a total of eight transformer blocks. We train 30 masked diffusion models, each on a different dataset constructed according to the same MRF procedure (Sec. 3.2), for 20,000 gradient steps using the AdamW optimizer (Loshchilov & Hutter, 2019) with a learning rate of $10^{-3}$. The training objective follows that of LLaDA (Nie et al., 2025).

Dependency scores $s_{ij}$ are computed by averaging attention scores from the final two of the eight transformer blocks. We empirically observe similar patterns when using attention from one, two, or three of the final blocks.

**Evaluation details.** When evaluating the validity of attention scores for MRF edge and degree estimation, we use step-by-step decoding. Out of the nine decoding steps, metrics are computed up to the seventh step, as beyond this point only a single edge connecting two nodes remains. All results are obtained by averaging metrics over 100 random sampling paths and across 30 independently trained models. See Tab. 9 for details.

*Table 10.* Layer-selection ablation for attention-based dependency estimation on the synthetic MRF dataset.

| Layer Selection | AUC $\uparrow$ | Ratio (Edge/Non-edge) $\uparrow$ | OVR $\downarrow$ |
|---|---|---|---|
| Last 2 layers | **0.93** | **2.20** | **0.04** |
| Last 1 layer | 0.92 | 2.16 | 0.06 |
| Last 4 layers | 0.90 | 2.06 | 0.08 |
| All layers | 0.87 | 1.95 | 0.15 |
| First 4 layers | 0.81 | 1.81 | 0.22 |
| First 2 layers | 0.78 | 1.72 | 0.29 |
| First 1 layer | 0.73 | 1.62 | 0.37 |

**Layer-Selection Ablation.** We examine how the choice of attention layers affects the quality of the induced dependency graph. For each layer-selection scheme, we compute the attention-based edge scores and evaluate them against the ground-truth MRF structure using the same metrics as in Sec. 3.2. As shown in Tab. 10, the last two layers provide the most reliable dependency signal, achieving the highest AUC and edge/non-edge attention ratio and the lowest OVR. Nearby choices using later layers also perform competitively, whereas early layers substantially degrade all metrics. These results identifies the last two layers as the most reliable attention source in the eight-layer synthetic model, supporting the use of upper-layer attention. We therefore apply the same principle in the large-model setting by averaging attention over the final 30% of layers for dependency estimation in Sec. 4.3.

## C. Details and Additional Results on Analysis

Here we provide details of the experimental setup for Sec. 6, which examines the decoding behavior of various parallel decoding strategies on independently aggregated queries. We also include additional qualitative visualizations of the unmasking trajectories for each decoding method.

**Dataset construction.**  We curate the dataset by sampling from TriviaQA (Joshi et al., 2017). We filter out lengthy or ambiguous questions to retain simple, common-knowledge queries with concise answers.

To ensure high-quality factual recall, we applied several structural and semantic filters to the curated subset. All questions were required to terminate with a question mark or initiate with standard interrogative tokens (e.g., What, Which, Who, Where, When). We restricted queries to a single sentence with a maximum length of 120 characters. Furthermore, we mitigated ambiguity by explicitly excluding questions containing comparative or restrictive phrases—such as "which of the following," "best describes," or "most likely"—thereby ensuring a mapping to singular ground-truth entities. Finally, we removed questions beginning with "Why" or "How", as these typically necessitate complex generative explanations rather than the direct, noun-centric factual recall targeted in this study.

Below are examples of the filtered questions:

- *What is the name of the parson mentioned in the lyrics of the Christmas carol "Winter Wonderland"?*

- *Who played 'Peter Pan' in Spielberg's 'Hook'?*

- *Who won the Formula 1 World Championship in 1992?*

- *Who wrote, produced and directed epic film, Avatar?*

After filtering, we obtain a total of 12,380 questions. We then randomly sample five independent questions and aggregate them into a single instance, constructing 500 samples in total (2,500 individual questions). To encourage explicit reasoning, we include instructions and illustrative examples that require the model to explain its reasoning rather than output only final answers. The prompt template is shown below.

```
Please answer the following 5 independent questions.

Instructions:
1) Write answers in full sentence format.
2) Explain your reasoning step by step before stating the final answer.
3) Write the final answer after "Therefore, the answer is".
4) Keep the numbering format [1], [2], [3], [4], [5].

Here are examples showing how your answers should be written:

[Example 1]
Question: What is the smallest country in the world by area?
Answer: To determine the smallest country, we compare the land areas of sovereign states.
Although countries like Monaco and San Marino are very small, Vatican City has an area
of about 0.44 square kilometers, which is smaller than any other country.
Therefore, the answer is Vatican City.

[Example 2]
Question: How many continents are there on Earth?
Answer: Continents are large continuous landmasses. These include Africa, Antarctica,
Asia, Europe, North America, South America, and Australia.
Counting these gives a total of seven continents.
Therefore, the answer is seven.

Now answer the following questions in the same style:

[1] {Question 1}
[2] {Question 2}
[3] {Question 3}
[4] {Question 4}
[5] {Question 5}
```

**Evaluation details.**  We evaluate accuracy by string matching against ground truth answers and aliases in the model outputs. For each prompt instance, which contains five independent questions, we first extract the corresponding answer segment using the bracketed markers [1], [2], etc. We then identify the predicted answer by parsing the text following

the canonical phrase "Therefore, the answer is" (or the fallback "The answer is") and perform case-insensitive substring matching against the ground truth answer and its alias set. If no such phrase is detected, we fall back to matching over the entire generated segment. We report accuracy as the fraction of questions for which the answer (or an alias) is matched, aggregated over all questions in the dataset. In addition, we report the average number of function evaluations (NFE) as a measure of decoding steps (Tab. 2), computed by averaging the per-sample NFE values in each method's output file.

**Hyperparameter details.**    All experiments are conducted using LLaDA. We sweep several hyperparameters for each baseline. For EB-Sampler, we vary the parameter $\gamma$ over $\{0.05, 0.1, 0.15, 0.2\}$, while for Fast-dLLM and KLASS, we sweep the confidence threshold over $\{0.6, 0.7, 0.8, 0.9\}$. For KLASS, we sweep the KL threshold over $\{0.001, 0.0015, 0.015, 0.1, 0.2\}$. We report results for each baseline using the hyperparameter setting that yields the highest accuracy. For DAPD, we apply a linear $\tau_t$ schedule over $[0.01, 0.05]$, consistent with the default configuration used in our LLaDA experiments.

**Additional qualitative results.**    Visualizations of the unmasking trajectories for EB-Sampler and KLASS, corresponding to the analysis in Fig. 1, are provided in Figs. 9 and 11, respectively. We also include examples for all compared methods: DAPD (Figs. 7 and 8), EB-Sampler (Figs. 9 and 10), KLASS (Figs. 11 and 12), and Fast-dLLM (Figs. 13 and 14).

# D. Related Work, Discussion and Limitations

**Concurrent Related Work.**    Concurrent work has also studied token dependencies in diffusion language models. Li et al. (2026) identify the factorization barrier caused by fully factorized token-wise outputs and address it by augmenting the model with a tractable probabilistic inference layer, whereas DAPD keeps the pretrained dLLM fixed and mitigates joint–marginal mismatch at inference time.

Most closely related, Luo et al. (2026) construct an attention-based dependency graph for training-free fast inference. However, DAWN primarily uses this graph to relax confidence thresholds: high-confidence committed tokens act as anchors, and masked positions strongly coupled to these anchors can be decoded under a lower confidence threshold; it also avoids jointly decoding strongly coupled low-confidence candidates. In contrast, DAPD uses attention to build a dependency graph directly over masked positions, independent of confidence relaxation, and selects an independent set as the parallel decoding batch. This makes the graph structure align more directly with the joint–marginal mismatch problem and motivates our degree-prioritized scheduling rule for simplifying the residual graph across decoding steps.

**Role of the MRF Formulation.**    Although the MRF formulation may appear to suggest exact probabilistic inference, this is not its intended role in DAPD. Rather, its role is to turn dependency-aware parallel decoding into a concrete graph-selection problem. By representing masked positions as nodes and strong attention-based interactions as edges, the MRF provides a conflict graph: edges indicate pairs that should not be co-updated, while independent sets define candidate parallel decoding batches. This graph view is algorithmically useful because decoding a set of tokens removes nodes and changes the structure of future decisions.

DAPD therefore does not merely seek a large independent set myopically at each step. Instead, it aims to reduce the total number of decoding steps by simplifying the residual graph over time, which naturally connects the method to dynamic graph coloring. This also motivates the Welsh–Powell-style degree-prioritized rule: resolving high-degree nodes early removes highly connected conflicts and makes the remaining graph easier to decode in parallel. Since exact conditional-independence testing is infeasible during dLLM decoding, DAPD uses attention-derived edge scores as a lightweight proxy and avoids co-updating positions with strong direct interactions, the most likely source of joint–marginal mismatch.

**Independence and Decoding Priority.**    Dependency-aware selection is intended to enable safe parallelism, not to replace confidence as a token-quality measure. In sequential decoding, confidence-based ordering is often effective because only one token is updated at a time. In parallel decoding, however, confident but mutually dependent positions can still cause joint–marginal mismatch when decoded together. Confidence-only methods avoid such collisions through conservative thresholds, but this misses many weakly dependent positions that could be safely updated in parallel. DAPD separates token reliability from cross-position dependence: confidence measures the former, while the attention-induced graph estimates the latter. This allows DAPD to select larger low-conflict batches and reduce decoding steps more aggressively than confidence-only strategies.

**Limitations.**   DAPD relies on heuristic choices such as the threshold schedule for graph construction and the attention layers used for dependency estimation. Although our ablations indicate that later-layer attention provides a reliable signal in our settings, more principled adaptive thresholding and layer-aggregation strategies may improve robustness across architectures and tasks. In addition, attention-based edge scores should be viewed as a practical proxy for conditional dependence rather than exact conditional-independence tests. This limitation motivates future work on more principled dependency estimators for dLLM decoding.

Questions

[1] What is the capital of North Korea?

[2] Who wrote the poem whose lengthy title is usually shortened to "The Tale of Tsar Saltan"?

[3] Who wrote the narrative poem The Rape of the Lock between 1712 and 1717?

[4] Which planet has the strongest gravity in our solar system?

[5] In which year was the Battle of Balaclava (including the Charge of the Light Brigade)?

## DAPD

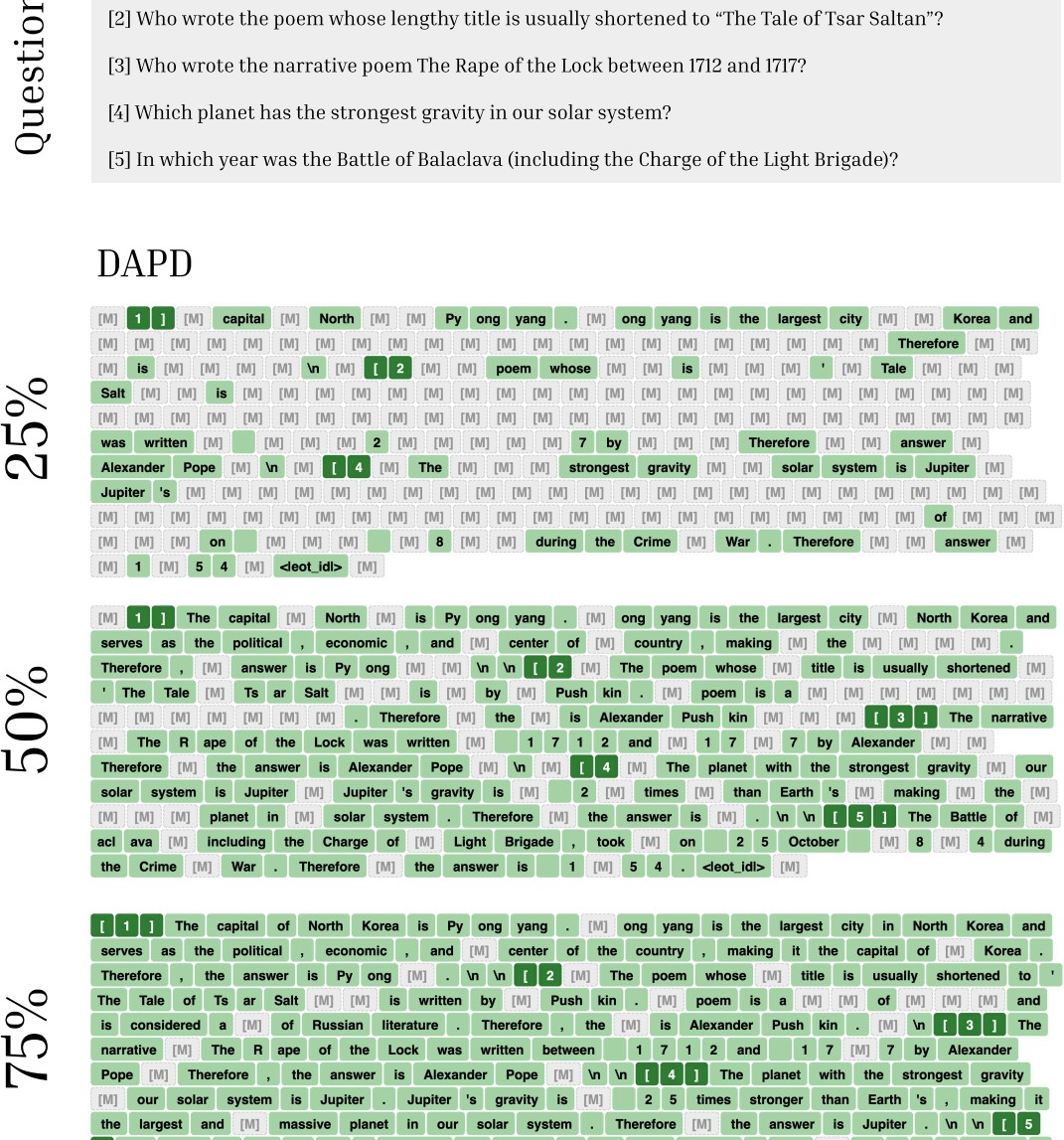

*Figure 7.* Illustrative example of DAPD. Five independent questions (top) and their unmasked states across decoding steps.

Questions

[1] In which country is Keflavík Airport?

[2] In what year did Tony Blair become leader of the Labour Party?

[3] What is the name of Janet Jackson's deceased superstar brother?

[4] In which year did Barbados join the Commonwealth?

[5] In which year did Franco capture Madrid and win the Spanish Civil War?

## DAPD

*Figure 8.* Illustrative example of DAPD. Five independent questions (top) and their unmasked states across decoding steps.

Questions

[1] What is the capital of North Korea?

[2] Who wrote the poem whose lengthy title is usually shortened to "The Tale of Tsar Saltan"?

[3] Who wrote the narrative poem The Rape of the Lock between 1712 and 1717?

[4] Which planet has the strongest gravity in our solar system?

[5] In which year was the Battle of Balaclava (including the Charge of the Light Brigade)?

## EB-Sampler

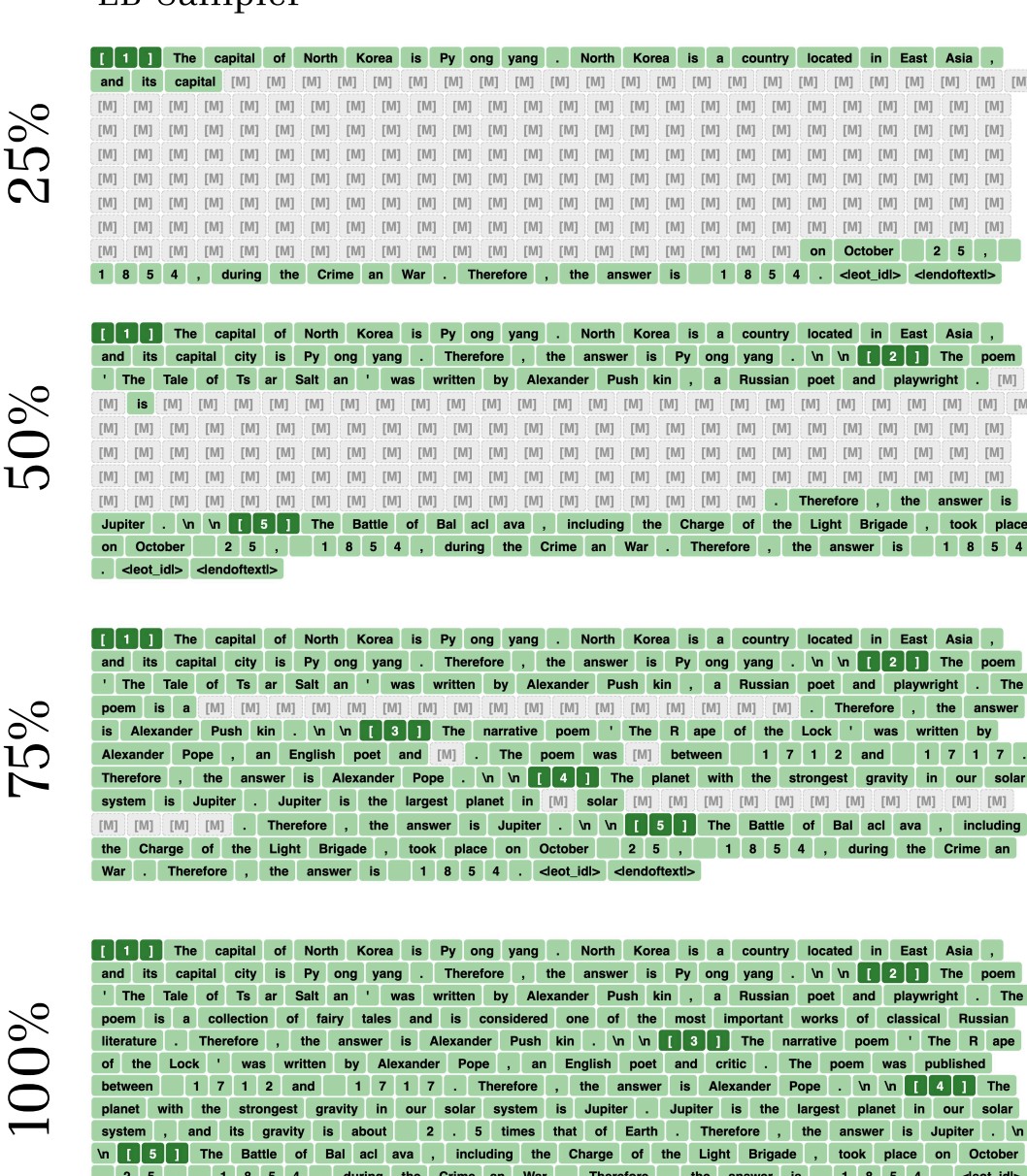

*Figure 9.* Illustrative example of EB-Sampler. Five independent questions (top) and their unmasked states across decoding steps.

**Questions**

[1] In which country is Keflavík Airport?

[2] In what year did Tony Blair become leader of the Labour Party?

[3] What is the name of Janet Jackson's deceased superstar brother?

[4] In which year did Barbados join the Commonwealth?

[5] In which year did Franco capture Madrid and win the Spanish Civil War?

## EB-Sampler

**25%**

[ 1 ] Ke pl av ik Airport is located in Iceland . Ke fl av ik is a town in the [M] part of Iceland , and the airport serves as the [M] airport [M] the [M] . Therefore , the answer is Iceland . \n \n [ 2 ] Tony Blair became [M] [M] [M] [M] [M] [M] [M] [M] [M] [M] [M] [M] [M] [M] [M] [M] [M] [M] [M] ... [M] . Therefore , the answer is 1 9 3 9 . <|eot_id|> <|endoftext|>

**50%**

[ 1 ] Ke pl av ik Airport is located in Iceland . Ke fl av ik is a town in the [M] part of Iceland , and the airport serves as the primary airport for the region . Therefore , the answer is Iceland . \n \n [ 2 ] Tony Blair became the leader of the Labour Party in 1 9 9 4 . He was elected as the leader of the Labour Party in the United Kingdom in 1 9 9 4 . Therefore , the answer is 1 9 9 4 . \n \n [ 3 ] Janet Jackson [M] deceased [M] ... [M] . Therefore , the answer is 1 9 3 9 . <|eot_id|> <|endoftext|>

**75%**

[ 1 ] Ke pl av ik Airport is located in Iceland . Ke fl av ik is a town in the [M] part of Iceland , and the airport serves as the primary airport for the region . Therefore , the answer is Iceland . \n \n [ 2 ] Tony Blair became the leader of the Labour Party in 1 9 9 4 . He was elected as the leader of the Labour Party in the United Kingdom in 1 9 9 4 . Therefore , the answer is 1 9 9 4 . \n \n [ 3 ] Janet Jackson 's deceased superstar brother is Michael Jackson . Michael Jackson was a [M] [M] singer , dancer , and record producer who passed away in 2 0 0 9 . Therefore , the answer is Michael Jackson . \n \n [ 4 ] Barbados joined [M] Commonwealth in 1 9 6 6 . Barbados gained independence from [M] ... [M] in 1 9 3 9 . Therefore , the answer is 1 9 3 9 . <|eot_id|> <|endoftext|>

**100%**

[ 1 ] Ke pl av ik Airport is located in Iceland . Ke fl av ik is a town in the southwestern part of Iceland , and the airport serves as the primary airport for the region . Therefore , the answer is Iceland . \n \n [ 2 ] Tony Blair became the leader of the Labour Party in 1 9 9 4 . He was elected as the leader of the Labour Party in the United Kingdom in 1 9 9 4 . Therefore , the answer is 1 9 9 4 . \n \n [ 3 ] Janet Jackson 's deceased superstar brother is Michael Jackson . Michael Jackson was a world -renowned singer , dancer , and record producer who passed away in 2 0 0 9 . Therefore , the answer is Michael Jackson . \n \n [ 4 ] Barbados joined the Commonwealth in 1 9 6 6 . Barbados gained independence from the United Kingdom in 1 9 6 6 and became a member of the Commonwealth of Nations . Therefore , the answer is 1 9 6 6 . \n \n [ 5 ] Franco captured Madrid and won the Spanish Civil War in 1 9 3 9 . The Spanish Civil War began in 1 9 3 6 and ended with the capture of Madrid by Franco in 1 9 3 9 . Therefore , the answer is 1 9 3 9 . <|eot_id|> <|endoftext|>

*Figure 10.* Illustrative example of EB-Sampler. Five independent questions (top) and their unmasked states across decoding steps.

**Questions**

[1] What is the capital of North Korea?

[2] Who wrote the poem whose lengthy title is usually shortened to "The Tale of Tsar Saltan"?

[3] Who wrote the narrative poem The Rape of the Lock between 1712 and 1717?

[4] Which planet has the strongest gravity in our solar system?

[5] In which year was the Battle of Balaclava (including the Charge of the Light Brigade)?

## KLASS

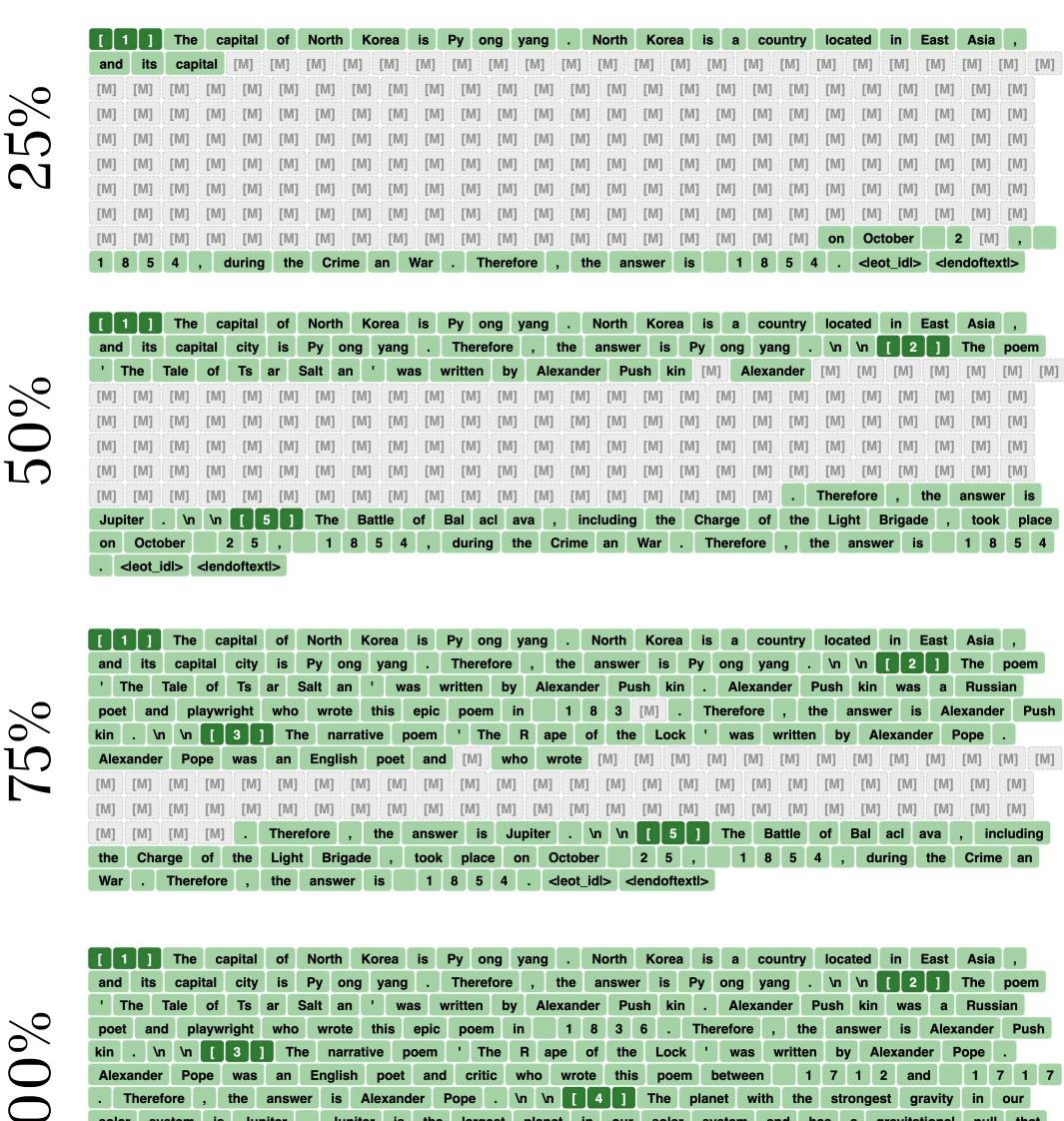

*Figure 11.* Illustrative example of KLASS. Five independent questions (top) and their unmasked states across decoding steps.

Questions

[1] In which country is Keflavík Airport?

[2] In what year did Tony Blair become leader of the Labour Party?

[3] What is the name of Janet Jackson's deceased superstar brother?

[4] In which year did Barbados join the Commonwealth?

[5] In which year did Franco capture Madrid and win the Spanish Civil War?

## KLASS

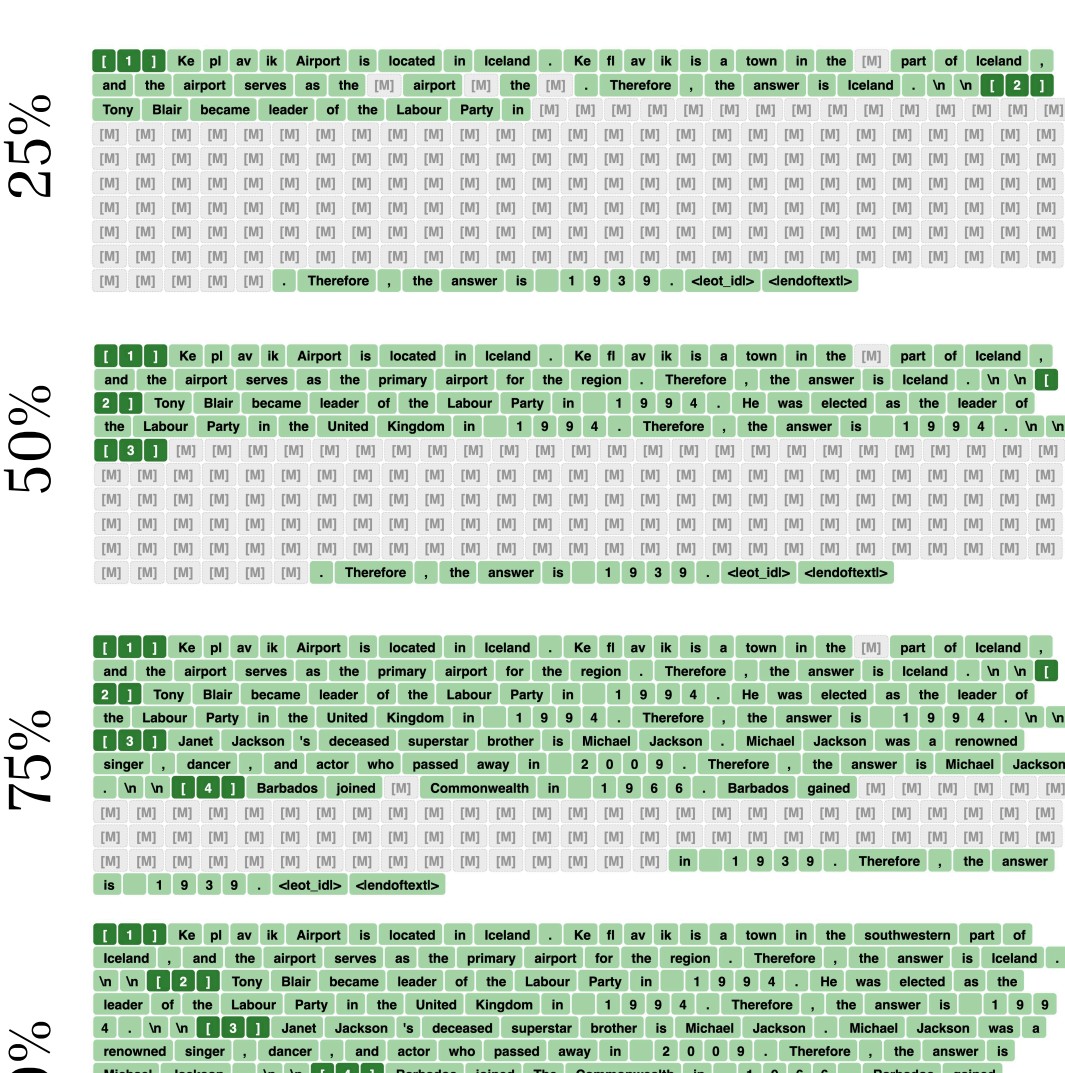

*Figure 12.* Illustrative example of KLASS. Five independent questions (top) and their unmasked states across decoding steps.

## Fast-dLLM

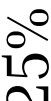
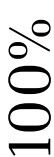
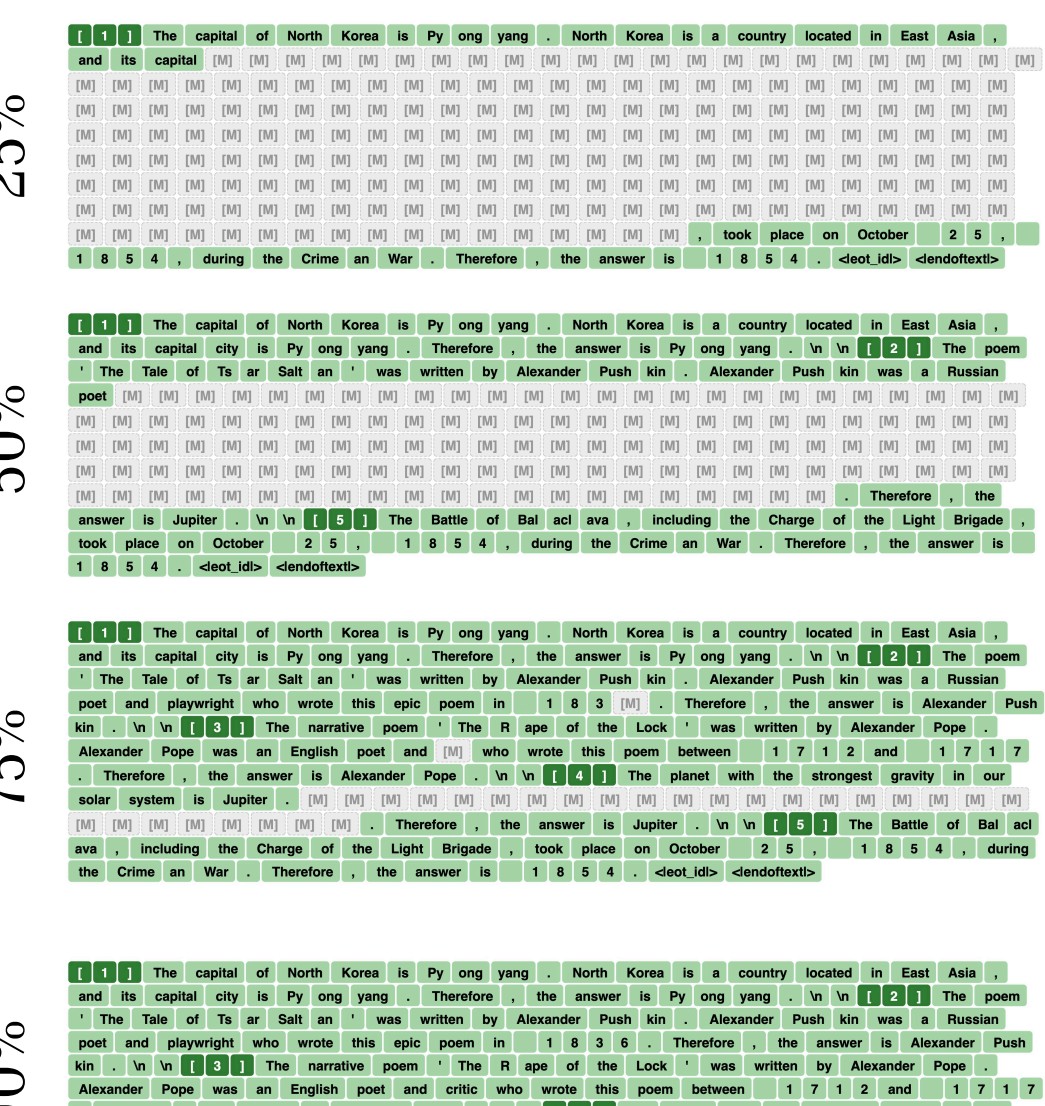

*Figure 13.* Illustrative example of Fast-dLLM. Five independent questions (top) and their unmasked states across decoding steps.

Questions

[1] In which country is Keflavík Airport?

[2] In what year did Tony Blair become leader of the Labour Party?

[3] What is the name of Janet Jackson's deceased superstar brother?

[4] In which year did Barbados join the Commonwealth?

[5] In which year did Franco capture Madrid and win the Spanish Civil War?

## Fast-dLLM

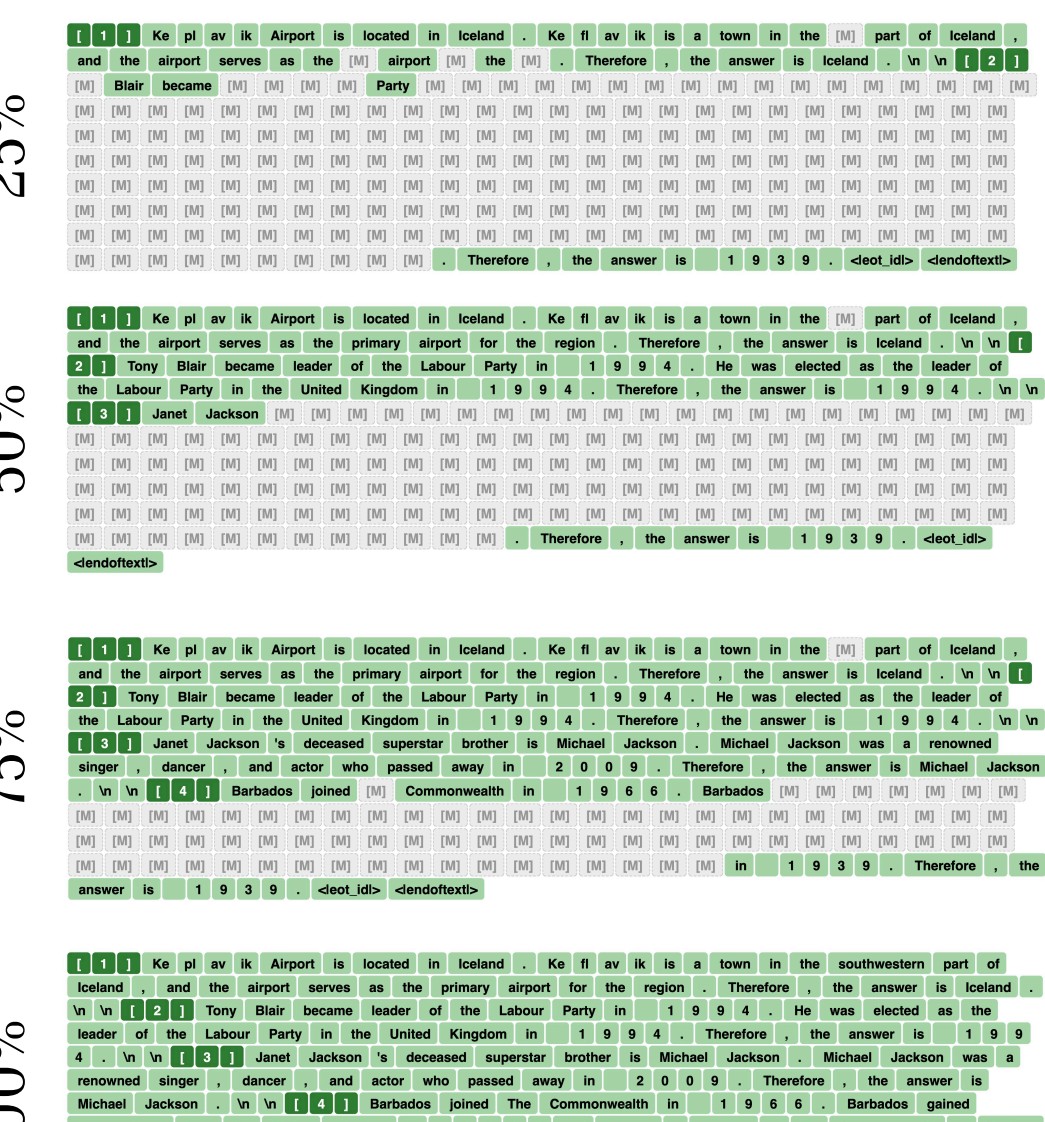

*Figure 14.* Illustrative example of Fast-dLLM. Five independent questions (top) and their unmasked states across decoding steps.

