# OpenReview forum: "DAPD: Dependency-Aware Parallel Decoding via Attention for Diffusion LLMs"
_ICML.cc/2026/Conference — ICML 2026 regular_

### Official Review · Reviewer_u6Zz · 2026-03-09

**Soundness:** 2
**Presentation:** 2
**Significance:** 3
**Originality:** 2
**Overall Recommendation:** 4
**Confidence:** 4

**Summary:**

This paper proposes DAPD, a dependency-aware parallel decoding method for diffusion language models (DLMs) that aims to improve decoding efficiency and generation quality. The idea is to estimate token dependencies using attention weights and construct a graph representing interactions between masked tokens. Inspired by Markov Random Field intuition, the method identifies groups of tokens that are approximately independent and decodes them simultaneously, prioritizing tokens with lower dependency degrees to reduce conflicts during parallel updates. This strategy encourages the model to first resolve independent regions across the sequence before filling remaining gaps.

**Compliance With Llm Reviewing Policy:**

Affirmed.

**Final Justification:**

I thank the authors for the response. I am keeping my positive score.

**Key Questions For Authors:**

1. Could the authors clarify the intuition or provide additional analysis on why prioritizing independence—rather than confidence or uncertainty—leads to better decoding outcomes?

2. Could the authors clarify what algorithmic benefit the MRF formulation provides beyond serving as a motivation?

**Limitations:**

See weaknesses.

**Strengths And Weaknesses:**

Strengths:

1. The paper addresses a limitation of diffusion language models: the inefficiency and instability of existing parallel decoding strategies. By explicitly modeling token dependencies and prioritizing updates in independent regions of the sequence, the method provides a principled attempt to improve both decoding efficiency and generation quality.

2. The paper introduces an interesting conceptual framework by interpreting token interactions through a dependency graph inspired by Markov Random Fields. This perspective offers an intuitive way to reason about which tokens can be updated simultaneously during decoding and provides a structured approach to selecting token groups for parallel updates.

Weaknesses:

1. The paper does not clearly justify why decoding tokens from independent regions earlier should improve generation quality or efficiency beyond enabling parallelism. Independence alone does not imply that a token should be decoded earlier than other high-confidence tokens.

2. The connection to Markov Random Fields is not fully convincing. While the paper introduces an MRF interpretation of token dependencies, the proposed decoding algorithm does not actually perform probabilistic inference on the MRF. Instead, the graph is used only as a heuristic structure for selecting tokens to decode. As a result, the role of the MRF formulation appears largely conceptual rather than algorithmically necessary.

3. The dependency graph is constructed using attention weights, but the paper does not clearly justify which layers’ attention should be used or why those layers best capture token dependencies relevant for decoding. Since attention patterns vary substantially across transformer layers, this design choice may significantly affect the resulting graph structure.

---

> ### Author Rebuttal · Authors · 2026-03-31
>
> ### **Response to W1 & Q1: Independence vs decoding priority**
>
> We thank the reviewer for this question. To clarify, independence is prioritized precisely to enable safe parallelism and drastically boost inference speed, rather than to directly improve generation quality.
>
> In standard sequential (step-by-step) decoding, confidence-based decoding is widely used and often yields better performance in practice. However, in parallel decoding, simultaneously unmasking confident but dependent tokens causes collisions that degrade output quality. Therefore, identifying mutually independent tokens is the absolute prerequisite for accelerating generation safely [1-4].
>
> Confidence-only baselines try to avoid these collisions using overly conservative probability thresholds (e.g., >0.9). This bottlenecks speed by missing many structurally independent tokens that could have been safely decoded together. By explicitly measuring dependency, DAPD safely maximizes the number of tokens updated per step, directly reducing the total number of decoding steps.
>
>
> ### **Response to W2 & Q2: Role of MRF formulation**
> We appreciate the reviewer for this insightful comment. While we agree that the MRF formulation is not used for exact probabilistic inference, its role goes well beyond conceptual motivation by providing the structural object on which our decoding algorithm operates. In particular, once masked positions are represented as nodes and strong inter-token conflicts are represented as edges, parallel decoding becomes a structured decision problem over the dependency graph. This is the key algorithmic benefit of introducing the graph view: without such a graph, one can assess only which tokens seem safe to decode together at the current step, but not how a current decoding choice shapes the structure of future decisions.
>
> More concretely, our objective is not merely to extract a large independent set myopically at each iteration, but to choose tokens whose removal most effectively simplifies the residual graph and thereby reduces the total number of future decoding steps (NFE). This is exactly why the problem becomes naturally connected to dynamic graph coloring: minimizing total decoding steps corresponds to covering the evolving graph with as few independent sets as possible.
>
> Under this view, our Welsh–Powell-motivated degree-prioritized rule is not an arbitrary heuristic. Prioritizing high-degree tokens is principled precisely because resolving highly connected conflicts early tends to make the remaining graph easier to decode in parallel later. We will revise the paper to make this algorithmic role of the MRF formulation more explicit.
> ### **Response to W3: Layer selection for dependency graph**
>
>
> We thank the reviewer for raising this point. We apologize for not stating this clearly: throughout the paper, we use the final 30% of layers, which in the toy setting corresponds to the last 2 layers. This choice is motivated by capturing dependencies between unknown masked tokens.
>
> Unlike standard attention (from known context to masked tokens), establishing dependencies between two unknown masked tokens requires the model to first process the surrounding context and update the masks' initial representations. Consequently, meaningful mask-to-mask interactions naturally emerge in the later layers.
>
> To validate this, we conducted ablation studies on our synthetic MRF dataset, where ground-truth dependencies are known. As shown below, the last 2 layers (the final 30%) provide the most reliable signal for recovering the dependency structure.
>
> |Layer Selection|AUC(↑)|Ratio(↑)|OVR(↓)|
> |-|-:|-:|-:|
> |**Last 2 (L7–8, 30%)**|**0.93**|**2.20**|**0.04**|
> |Last 1 (L8, 10%)|0.92|2.16|0.06|
> |Last 4 (L5–8, 50%)|0.90|2.06|0.08|
> |All layers (100%)|0.87|1.95|0.15|
> |First 4 (L1–4, 50%)|0.81|1.81|0.22|
> |First 2 (L1–2, 30%)|0.78|1.72|0.29|
> |First 1 (L1, 10%)|0.73|1.62|0.37|
>
> These results also suggest that using the latter portion of the model remains robust across nearby choices (e.g., the last 10% or 50%), while using only early layers struggles to capture reliable dependencies. Based on these findings, we consistently applied the 30% rule to our larger experiments (LLaDA and Dream). This configuration provides stable, generalizable performance across architectures without requiring task-specific tuning. We will include this clarification in the revision.
>
> ---
>
> [1] Azangulov et al., Parallel Sampling from Masked Diffusion Models via Conditional Independence Testing, ICLR, 2026.\
> [2] Wu et al., Fast-dLLM: Training-free Acceleration of Diffusion LLM by Enabling KV Cache and Parallel Decoding, ICLR, 2026.\
> [3] Ben-Hamu et al., Accelerated Sampling from Masked Diffusion Models via Entropy Bounded Unmasking, NeurIPS, 2025.\
> [4] Kang et al., ParallelBench: Understanding the Trade-offs of Parallel Decoding in Diffusion LLMs, ICLR, 2026.

---

> > ### Author Rebuttal · Reviewer_u6Zz · 2026-04-03
> >
> > I thank the authors for their detailed response. My concerns have been addressed, and I will maintain my positive score.

---

### Official Review · Reviewer_jweD · 2026-03-10

**Soundness:** 3
**Presentation:** 3
**Significance:** 3
**Originality:** 3
**Overall Recommendation:** 4
**Confidence:** 3

**Summary:**

The paper proposes a training-free decoding method that mitigates the joint-marginal mismatch. It leverages attention scores to construct a MRF for modeling dependencies among masked tokens and determining the unmasking schedule by selecting an independent set of the MRF. Experimental results demonstrate its good performance.

**Compliance With Llm Reviewing Policy:**

Affirmed.

**Final Justification:**

My concerns have been addressed. I will keep my positive score.

**Key Questions For Authors:**

NA

**Limitations:**

Yes

**Strengths And Weaknesses:**

The proposed idea is reasonable and novel. The techniques for implementing the idea are well-chosen/designed. The experimental results are good.

I'm a little bit unsure of the idea that two mutually dependent masked tokens shall always not be unmasked simultaneously. Suppose two masked tokens both have low entropy (i.e., they have high confidence in their decoding) and they have strong attention between them (e.g., perhaps they constitute a phrase like "New York"), shouldn't they be generated simultaneously?

In the LLaDA experiments, DAPD is evaluated in a single-block setting, while the baselines uses a 4-block setup with EOS suppression. As shown in Table 4, the 4-block setup requires more decoding steps. It would be nice to know how DAPD would perform with a 4-block setup. Would it achieve even better performance when given more steps?

Recently released DLMs, such as LLaDA 2.1, natively employ a block diffusion mechanism and hence a smaller block size. A smaller block size may lead to denser dependencies within a single block. Would that make DAPD less effectiveness?

Question about implementation details of DAPD: When the remaining mask ratio falls below 50% (hence a sparse MRF), DAPD switches to a confidence-threshold rule. If we continue to follow the greedy selection, we would also select most of the nodes that have a near-zero degree. Why is it necessary to switch to the confidence-threshold rule?

More related work:
- [Breaking the Factorization Barrier in Diffusion Language Models](https://arxiv.org/html/2603.00045v1)
- [DAWN: Dependency-Aware Fast Inference for Diffusion LLMs](https://arxiv.org/html/2602.06953v1)

---

> ### Author Rebuttal · Authors · 2026-03-31
>
> ### **Response to W1: Concurrent unmasking of high confidence tokens**
>
> We thank the reviewer for this insightful point. The core challenge in parallel decoding is that dLLMs learn conditional marginals $p_{\theta}(x^i | x_t)$, not the full joint distribution. Sampling multiple tokens simultaneously requires the approximation: $p_{\theta}(x^i, x^j | x_t) \approx p_{\theta}(x^i | x_t) p_{\theta}(x^j | x_t)$.
>
> The reviewer is correct that when tokens have high confidence, this approximation naturally holds. Even if tokens are seemingly dependent, they become effectively independent when their individual probabilities approach 1. Because both the true joint probability and the product of their marginals are close to 1, the probability mass concentrates. This explains why baseline confidence-thresholding (e.g., Fast-dLLM) works well in high-certainty regimes.
>
> However, relying exclusively on high confidence limits acceleration. DAPD’s key contribution is using self-attention to identify structurally independent tokens. This ensures the factorization $p_{\theta}(x^i, x^j | x_t) \approx p_{\theta}(x^i | x_t) p_{\theta}(x^j | x_t)$ holds even for low-confidence tokens, allowing us to safely unmask a much larger set of tokens in parallel without degrading performance.
>
> ### **Response to W2 & W3: Block setting and generalization**
>
> We appreciate the reviewer’s suggestion to evaluate DAPD across multi-block settings.
> To provide context on our evaluation choices, multi-block setups (such as the 4-block setting) are heavily utilized by baseline methods primarily because those methods suffer from severe EOS overflow and degenerate completely in a 1-block setting (as detailed in Table 4, Appendix A). For these baselines, moving from 1 block to 4 blocks is not an algorithmic improvement; it is a necessity just to achieve reasonable performance.
>
> In contrast, operating effectively in a 1-block setting is a distinct, inherent strength of DAPD. Conceptually, a 1-block setup is preferable for parallel decoding because it allows the model to maximally utilize global parallelism by sampling simultaneously across the entire output sequence. Introducing blocks manually limits this scope by partitioning the sequence into smaller segments, which inherently bottlenecks the generation steps and restricts the maximum possible speedup.
>
> However, we entirely agree that comparing DAPD in the baselines' favored multi-block setup further solidifies our results. As shown in the HumanEval results below, DAPD also generalizes to varying block sizes.
>
> |Method|#Blocks|Acc.(↑)|TPS(↑)|
> |-|-:|-:|-:|
> |**DAPD**|1|0.366|106.0|
> ||4|0.378|57.1|
> ||8|0.396|43.6|
> ||16|0.415|34.6|
> |Fast-dLLM|4|0.372|51.4|
> |EB Sampler|4|0.372|39.2|
> |KLASS|4|0.378|25.6|
>
> When evaluated in the exact same 4-block regime as the baselines, DAPD not only functions well but also outperforms them in generation speed (Tokens Per Second) while maintaining or exceeding their performance.
> Ultimately, while DAPD demonstrates strong generalization and superior performance in multi-block settings, its ability to maintain high quality in the unrestricted 1-block setting is what unlocks its massive advantage in speed.
>
>
> ### **Response to W4: Switching criterion**
>
> We sincerely apologize for the confusing phrasing in the manuscript.
> The text said that we (DAPD) “switch to” a confidence-threshold rule, whereas what we intended was that the confidence threshold is “switch on.”
> We will correct this wording in the revision.
>
> To clarify, once the mask ratio falls below 50%, DAPD does not abandon its dependency-aware selection.
> Rather, it augments it with a confidence threshold to decode more aggressively in parallel.
> By this stage, earlier decoding steps have already resolved most strong dependencies, and the reviewer is correct that many remaining nodes have near-zero degree.
> Because the residual dependency graph is sparse, tokens with confidence > 0.9 are very likely to be effectively independent from one another, making this threshold a safe way to accelerate decoding without undermining the core dependency-aware mechanism.
>
> This therefore serves as a safe but aggressive filter that maximizes parallel decoding in the later stages before the standard DAPD selection handles the rest.
>
>
> ### **Related Works**
> Thank you for pointing out these concurrent works [1, 2]. We appreciate the suggestion and will add both references to the revised manuscript and discuss their relationship to our work.
>
> ---
>
> [1] Li et al., Breaking the Factorization Barrier in Diffusion Language Models, arXiv, 2026.\
> [2] Luo et al., DAWN: Dependency-Aware Fast Inference for Diffusion LLMs, arXiv, 2026.

---

> > ### Author Rebuttal · Reviewer_jweD · 2026-04-02
> >
> > My concerns have been addressed.

---

### Official Review · Reviewer_gn8T · 2026-03-11

**Soundness:** 3
**Presentation:** 3
**Significance:** 3
**Originality:** 3
**Overall Recommendation:** 4
**Confidence:** 3

**Summary:**

This article introduces the Dependency-Aware Parallel Decoding (DAPD), whichi is a training-free decoding method that uses self-attention to induce a conditional dependect graph to accelerate the decoding process of Diffusion LLMs.

**Compliance With Llm Reviewing Policy:**

Affirmed.

**Final Justification:**

My concerns have been addressed and I would keep my positive score.

**Key Questions For Authors:**

Please refer to the weakness part.

**Limitations:**

No limitation is mentioned in the article. One practical limitation that deserves explicit discussion is scalability: DAPD constructs an attention-induced dependency graph over the remaining masked positions and recomputes it at every decoding step, which may introduce nontrivial overhead as the masked set grows. Since the experiments are limited to relatively short settings (maximum generation length 256) and do not report end-to-end latency, it remains unclear how well the method scales to longer sequences in practice.

**Strengths And Weaknesses:**

Strength:

1. DAPD jointly combine the attention information and the confidence score to address the core joint–marginal mismatch problem.

2. With the model of the parallel decoding, DAPD is able to accelerate the decoding process of dLLM while maintaining performance.

Weakness:

1. To compute the edge score, the article only uses the last two layers' attention which requires more experiments as evidence.

2. DAPD requires graph construction, thresholding, degree estimation, greedy independent-set selection, and graph recomputation at every step. This may introduce extra cost of inference. Meanwhile, the paper mainly measures efficiency in terms of steps / NFE rather than end-to-end wall-clock latency.

---

> ### Author Rebuttal · Authors · 2026-03-31
>
> ### **Response to W1: Attention layer choice for edge scoring**
> We thank the reviewer for raising this point. We apologize for not stating this clearly: throughout the paper, we use the final 30% of layers, which in the toy setting corresponds to the last 2 layers. This choice is motivated by capturing dependencies between unknown masked tokens.
>
> Unlike standard attention (from known context to masked tokens), establishing dependencies between two unknown masked tokens requires the model to first process the surrounding context and update the masks' initial representations. Consequently, meaningful mask-to-mask interactions naturally emerge in the later layers.
>
> To validate this, we conducted ablation studies on our synthetic MRF dataset, where ground-truth dependencies are known. As shown below, the last 2 layers (the final 30%) provide the most reliable signal for recovering the dependency structure.
>
> |Layer Selection|AUC(↑)|Ratio(↑)|OVR(↓)|
> |-|-:|-:|-:|
> |**Last 2 (L7–8, 30%)**|**0.93**|**2.20**|**0.04**|
> |Last 1 (L8, 10%)|0.92|2.16|0.06|
> |Last 4 (L5–8, 50%)|0.90|2.06|0.08|
> |All layers (100%)|0.87|1.95|0.15|
> |First 4 (L1–4, 50%)|0.81|1.81|0.22|
> |First 2 (L1–2, 30%)|0.78|1.72|0.29|
> |First 1 (L1, 10%)|0.73|1.62|0.37|
>
> These results also suggest that using the latter portion of the model remains robust across nearby choices (e.g., the last 10% or 50%), while using only early layers struggles to capture reliable dependencies. Based on these findings, we consistently applied the 30% rule to our larger experiments (LLaDA and Dream). This configuration provides stable, generalizable performance across architectures without requiring task-specific tuning. We will include this clarification in the revision.
>
> ### **Response to W2: Explicit wall-clock overhead**
>
> We thank the reviewer for highlighting the importance of measuring end to end latency and scalability beyond decoding steps.
> The computational overhead introduced by graph construction in DAPD is negligible relative to the cost of the model forward pass.
> Specifically, dependency scores are computed directly from self-attention weights, which are already produced during the forward pass, and therefore require no additional model evaluations.
> Moreover, the selection process relies on a lightweight greedy mechanism, ensuring that the overall procedure remains efficient and scales well with increasing sequence length.
>
> As shown in the table below on HumanEval, DAPD not only reduces the number of decoding steps, but also achieves substantially higher TPS (Tokens Per Second) than the baselines, indicating that its advantage translates into practical end-to-end efficiency with only minimal additional computation.
>
> |Method|Acc.(↑)|Steps(NFE)(↓)|TPS(↑)|
> |-|-:|-:|-:|
> |**DAPD**|**0.366**|**45.2**|**106.0**|
> |Fast-dLLM|0.372|92.1|51.4|
> |EB Sampler|0.372|110.4|39.2|
> |KLASS|0.378|149.4|25.6|
> |Vanilla LLaDA|0.390|256.0|20.4|
>
> We further examine whether this advantage persists for longer generations.
> Specifically, we measured TPS across varying generation lengths, $L \in \{256, 512, 1024\}$ on Humaneval (Code) and GSM8K (Math), as shown below.
>
> |Task|Length|Acc.(↑)|Steps(NFE)(↓)|TPS(↑)|
> |-|-:|-:|-:|-:|
> |HumanEval|256|0.366|45.2|106.0|
> ||512|0.329|55.9|106.6|
> ||1024|0.317|67.5|93.6|
> |GSM8K|256|0.707|48.2|33.9|
> ||512|0.728|68.4|39.0|
> ||1024|0.707|79.9|45.8|
>
> These results show that DAPD maintains a comparable level of TPS as sequence length increases, demonstrating effective scalability in practice.
>
> ### **Response to Limitation**
> As suggested by the reviewer, we will include a more explicit discussion of end to end latency and the scalability of DAPD for longer sequences.
> We will also clarify and better justify our layer selection strategy based on the final 30 percent of layers.
> We thank the reviewer again for this valuable suggestion, which will help strengthen the manuscript.

---

> > ### Author Rebuttal · Reviewer_gn8T · 2026-04-05
> >
> > My concerns have been addressed.

---

### Official Review · Reviewer_RYbW · 2026-03-12

**Soundness:** 4
**Presentation:** 4
**Significance:** 3
**Originality:** 3
**Overall Recommendation:** 5
**Confidence:** 3

**Summary:**

This paper presents a training-free decoding method for diffusion-based large language models (dLLMs) to improve the efficiency and quality of parallel token generation. The key challenge in unlocking the benefits of parallel decoding with dLLMs is "joint-marginal mismatch," that is, when tokens sampled simultaneously from marginal distributions are locally plausible but globally inconsistent. The authors propose a method to identify independent tokens among masked tokens based on the model's self-attention mechanism. The method constructs a Markov Random Field (MRF) where masked tokens are nodes and edges represent strong attention-based interactions. Parallel decoding is then framed as selecting an independent set — a group of tokens with weak mutual dependencies — that could be safely unmasked simultaneously. Experiments on LLaDA and Dream demonstrate that DAPD achieves a superior accuracy-steps trade-off compared to existing baselines like Fast-dLLM and EB-Sampler.

**Compliance With Llm Reviewing Policy:**

Affirmed.

**Final Justification:**

I thank the authors for the clarifications. I keep my score (Accept).

**Key Questions For Authors:**

1. How sensitive is the performance to the number of layers used for attention averaging? Did you test using all layers or middle layers only?
2. Can you provide the actual wall-clock time comparisons for a full generation to see how the graph-processing overhead compares to the GPU forward pass time?

**Limitations:**

I would appreciate it if the authors discussed the limitations (if any) of their approach.

**Strengths And Weaknesses:**

The paper is sound, grounding its heuristic in the probabilistic framework of MRFs and the pairwise Markov property. The authors provide a theoretical justification relating low attention weights to approximate conditional independence. This link is empirically supported by toy experiments where the model effectively recovers the ground-truth dependency structure of a synthetic dataset.

## Strengths

- The method is training-free. It requires no auxiliary models, retraining, or modifications to the model architecture, making it highly portable.
- Unlike confidence-based baselines that often unmask tokens in contiguous blocks (resembling two-sided autoregressive decoding), DAPD produces a spatially dispersed unmasking pattern.
- It significantly reduces the number of function evaluations (NFE) by maximizing the size of safe parallel batches, reaching a significant  speedup over standard step-by-step decoding on TriviaQA tasks.

## Weaknesses

- (Minor) The method relies on a threshold $\tau_t$ and a linear schedule to define edges, which may require task-specific tuning.
- (Minor) The dependency scores are derived from averaging attention over the top two layers. While justified by the tendency of upper layers to capture global dependencies, the optimal choice of layers might vary across different architectures or model scales.
- The paper mentions that the goal is to minimize NFE, but it does not explicitly quantify the wall-clock overhead introduced by constructing the MRF graph at each step.

## Presentation

The paper is well-organized and the technical concepts are clearly illustrated. The problem is well motivated by a clear intuitive example.


## Significance

The paper addresses the important issue of inference throughput and provides a way to make dLLMs more competitive with established autoregressive models. The shift from local/sequential unmasking to global/distributed unmasking is a significant conceptual step in defining the "natural" decoding paradigm for non-sequential models.

## Originality

The originality lies in the application of classical graph coloring and independent set theory to the problem of diffusion decoding. While using attention as a proxy for importance is common, using it specifically to induce an MRF for the purpose of scheduling parallel token updates is a contribution to the field.

---

> ### Author Rebuttal · Authors · 2026-03-31
>
> ### **Response to W1: Linear tau-scheduling**
> We thank the reviewer for this important question regarding hyperparameters.
> While DAPD introduces the threshold hyperparameter $τ$ that may vary across tasks or architectures, we find that the method is generally robust and does not require extensive task specific tuning.
>
> We determined the initial value $τ_{\min}$ by analyzing the internal attention distributions of the models (LLaDA, Dream), as detailed in Appendix B.
> Then, we used $τ_{\max}$ with minor adjustments and observed that strong performance is achieved without heavy tuning.
>
> Furthermore, to ensure a fair comparison, we conducted comprehensive hyperparameter sweeps for all baseline algorithms to prevent advantages toward our method.
> These results suggest that while optimal tuning is possible, the dependency aware mechanism remains effective and generalizable across a wide range of settings.
>
> ### **Response to W2 & Q1: Layer choice justification**
> We thank the reviewer for raising this point. We apologize for not stating this clearly: throughout the paper, we use the final 30% of layers, which in the toy setting corresponds to the last 2 layers. This choice is motivated by capturing dependencies between unknown masked tokens.
>
> Unlike standard attention (from known context to masked tokens), establishing dependencies between two unknown masked tokens requires the model to first process the surrounding context and update the masks' initial representations. Consequently, meaningful mask-to-mask interactions naturally emerge in the later layers.
>
> To validate this, we conducted ablation studies on our synthetic MRF dataset, where ground-truth dependencies are known. As shown below, the last 2 layers (the final 30%) provide the most reliable signal for recovering the dependency structure.
>
> |Layer Selection|AUC(↑)|Ratio(↑)|OVR(↓)|
> |-|-:|-:|-:|
> |**Last 2 (L7–8, 30%)**|**0.93**|**2.20**|**0.04**|
> |Last 1 (L8, 10%)|0.92|2.16|0.06|
> |Last 4 (L5–8, 50%)|0.90|2.06|0.08|
> |All layers (100%)|0.87|1.95|0.15|
> |First 4 (L1–4, 50%)|0.81|1.81|0.22|
> |First 2 (L1–2, 30%)|0.78|1.72|0.29|
> |First 1 (L1, 10%)|0.73|1.62|0.37|
>
> These results also suggest that using the latter portion of the model remains robust across nearby choices (e.g., the last 10% or 50%), while using only early layers struggles to capture reliable dependencies. Based on these findings, we consistently applied the 30% rule to our larger experiments (LLaDA and Dream). This configuration provides stable, generalizable performance across architectures without requiring task-specific tuning. We will include this clarification in the revision.
>
> ### **Response to W3 & Q2: Explicit wall-clock overhead**
>
> We thank the reviewer for highlighting the importance of measuring latency and scalability.
> The computational overhead introduced by graph construction in DAPD is negligible relative to the cost of the model forward pass.
> Specifically, dependency scores are computed directly from self-attention weights, which are already produced during the forward pass, and therefore require no additional model evaluations.
> Moreover, the selection process relies on a lightweight greedy mechanism, ensuring that the overall procedure remains efficient and scales well with increasing sequence length.
>
> As shown in the table below on HumanEval, DAPD not only reduces the number of decoding steps, but also achieves substantially higher TPS (Tokens Per Second) than the baselines, indicating that its advantage translates into practical end-to-end efficiency with only minimal additional computation.
>
> |Method|Acc.(↑)|Steps(NFE)(↓)|TPS(↑)|
> |-|-:|-:|-:|
> |**DAPD**|**0.366**|**45.2**|**106.0**|
> |Fast-dLLM|0.372|92.1|51.4|
> |EB Sampler|0.372|110.4|39.2|
> |KLASS|0.378|149.4|25.6|
> |Vanilla LLaDA|0.390|256.0|20.4|
>
> We further examine whether this advantage persists for longer generations.
> Specifically, we measure TPS across varying generation lengths, $L \in \{256, 512, 1024\}$ on Humaneval and GSM8K, as shown below.
>
> |Task|Length|Acc.(↑)|Steps(NFE)(↓)|TPS(↑)|
> |-|-:|-:|-:|-:|
> |HumanEval|256|0.366|45.2|106.0|
> ||512|0.329|55.9|106.6|
> ||1024|0.317|67.5|93.6|
> |GSM8K|256|0.707|48.2|33.9|
> ||512|0.728|68.4|39.0|
> ||1024|0.707|79.9|45.8|
>
> These results show that DAPD maintains a comparable level of TPS as sequence length increases, demonstrating effective scalability in practice.
>
> ### **Response to Limitation**
> We acknowledge that the current choices of $\tau$ scheduling and layer selection are still heuristic. More principled strategies for these components may further improve the method, and we will add an explicit discussion of this limitation in the revision.

---

> > ### Author Rebuttal · Reviewer_RYbW · 2026-04-02
> >
> > I thank the authors for the clarifications. I keep my score (Accept).

---

### Decision · Program_Chairs · 2026-04-30

**Decision:**

Accept (regular)

**Comment:**

This paper proposes Dependency-Aware Parallel Decoding (DAPD), a training-free method that uses self-attention to identify mutually independent tokens for safe parallel unmasking in Diffusion LLMs. The authors successfully addressed reviewer concerns regarding wall-clock overhead, layer selection heuristics, and the role of the Markov Random Field framework by providing end-to-end latency benchmarks and synthetic ablation studies. By effectively resolving the joint-marginal mismatch problem without auxiliary models, DAPD achieves superior speedups and accuracy trade-offs across multiple benchmarks. The AC recommends the acceptance of this paper.